# HIV-1 Protease as DNA Immunogen against Drug Resistance in HIV-1 Infection: DNA Immunization with Drug Resistant HIV-1 Protease Protects Mice from Challenge with Protease-Expressing Cells

**DOI:** 10.3390/cancers15010238

**Published:** 2022-12-30

**Authors:** Stefan Petkov, Athina Kilpeläinen, Ekaterina Bayurova, Anastasia Latanova, Dzeina Mezale, Ilse Fridrihsone, Elizaveta Starodubova, Juris Jansons, Alesja Dudorova, Ilya Gordeychuk, Britta Wahren, Maria Isaguliants

**Affiliations:** 1Department of Microbiology, Tumor and Cell Biology, Karolinska Institutet, 171 65 Stockholm, Sweden; 2Department of Research, Riga Stradins University, LV-1007 Riga, Latvia; 3Chumakov Federal Scientific Center for Research and Development of Immune and Biological Products of Russian Academy of Sciences, 108819 Moscow, Russia; 4Engelhardt Institute of Molecular Biology, Russian Academy of Sciences, 119991 Moscow, Russia; 5Latvian Research and Study Centre, LV-1067 Riga, Latvia; 6Paul Stradins University Hospital, LV-1002 Riga, Latvia

**Keywords:** HIV-1, drug resistance, protease, DNA immunogen, CD8+ T-cell response, mammary gland adenocarcinoma 4T1luc2 cells, HIV-1-protein-expressing tumors, tumor growth, metastatic activity, protection from tumor challenge

## Abstract

**Simple Summary:**

DNA immunization with drug-resistant HIV-1 protease (DR PR) is proposed as immunotherapy to prevent the evolution of HIV-1 towards drug resistance and reduce the number of infected cells producing drug-resistant virus. We designed a consensus gene of PR of HIV-1 clade A FSU_A strain which infects millions of people in the territory of the former Soviet Union and which has so far remained highly conserved due to rapid transfer within high-risk groups of the population. Into the synthetic gene of the consensus PR, we introduced DR mutations common to this strain plus a PR-inactivating mutation, and characterized the immunogenic profiles of DR-PR-encoding plasmids in mice. Finally, we tested if DNA immunization with DR PR protected mice against challenge with murine adenocarcinoma cells designed to express a DR PR variant. Using this model, we demonstrated that the immune response against DR PR protects mice against the growth and metastatic activity of tumor cells expressing the respective DR PR variant. Protection relies on a cytolytic T-cell response against single epitopes harboring the DR mutation. This is a proof of concept that DNA immunization can induce a cytolytic immune response recognizing single amino acids including DR mutations, promoting the application of therapeutic DNA vaccines against DR HIV-1.

**Abstract:**

DNA immunization with HIV-1 protease (PR) is advanced for immunotherapy of HIV-1 infection to reduce the number of infected cells producing drug-resistant virus. A consensus PR of the HIV-1 FSU_A strain was designed, expression-optimized, inactivated (D25N), and supplemented with drug resistance (DR) mutations M46I, I54V, and V82A common for FSU_A. PR variants with D25N/M46I/I54V (PR_Ai2mut) and with D25N/M46I/I54V/V82A (PR_Ai3mut) were cloned into the DNA vaccine vector pVAX1, and PR_Ai3mut, into a lentiviral vector for the transduction of murine mammary adenocarcinoma cells expressing luciferase 4T1luc2. BALB/c mice were DNA-immunized by intradermal injections of PR_Ai, PR_Ai2mut, PR_Ai3mut, vector pVAX1, or PBS with electroporation. All PR variants induced specific CD8+ T-cell responses revealed after splenocyte stimulation with PR-derived peptides. Splenocytes of mice DNA-immunized with PR_Ai and PR_Ai2mut were not activated by peptides carrying V82A, whereas splenocytes of PR_Ai3mut-immunized mice recognized both peptides with and without V82A mutation. Mutations M46I and I54V were immunologically silent. In the challenge study, DNA immunization with PR_Ai3mut protected mice from the outgrowth of subcutaneously implanted adenocarcinoma 4T1luc2 cells expressing PR_Ai3mut; a tumor was formed only in 1/10 implantation sites and no metastases were detected. Immunizations with other PR variants were not protective; all mice formed tumors and multiple metastasis in the lungs, liver, and spleen. CD8+ cells of PR_Ai3mut DNA-immunized mice exhibited strong IFN-γ/IL-2 responses against PR peptides, while the splenocytes of mice in other groups were nonresponsive. Thus, immunization with a DNA plasmid encoding inactive HIV-1 protease with DR mutations suppressed the growth and metastatic activity of tumor cells expressing PR identical to the one encoded by the immunogen. This demonstrates the capacity of T-cell response induced by DNA immunization to recognize single DR mutations, and supports the concept of the development of immunotherapies against drug resistance in HIV-1 infection. It also suggests that HIV-1-infected patients developing drug resistance may have a reduced natural immune response against DR HIV-1 mutations causing an immune escape.

## 1. Introduction

Since the mid-1990s, countries of the former Soviet Union (FSU) have experienced a rapidly developing HIV-1 epidemic in contrast to the global decrease in the number of new HIV-1 cases, specifically in Europe [1,2,3,4,5,6]. The epidemic has been initially driven by injection drug use (IDU), but has spread into the general population as manifested by the increasing number of cases of sexual transmission [7]. The predominant strain is still a subtype A variant of low genetic diversity known as IDU-A or FSU-A, first identified in 1996 in Ukraine [4,5,6]. 

Highly active antiretroviral therapy (ART) provided by the healthcare systems helps to decrease viral load and spread. However, due to the lack of proofreading activity of viral reverse transcriptase, mutations occur which make the virus less susceptible or insusceptible to drugs [8]. This variability, in combination with the selective pressure imposed by ART, leads to the generation of drug-resistant (DR) HIV-1 variants, spreading due to poor compliance to therapy, and eventual drug shortages [9,10,11]. The circulation of such DR HIV-1 strains considerably limits therapeutic options.

Nucleic-acid-based mRNA and DNA vaccines have shown promising results in stimulating strong cellular and humoral immune responses against COVID-19; plus, the rapid and easy production process makes them specifically attractive compared to other vaccine platforms [12]. Their primary advantage for targeting HIV-1 is in the ability to induce the local expression of target antigens and subsequently trigger an antigen-specific cellular immune response by CD8+ and CD4+ T cells [13]. Both SIV studies and clinical HIV-1 trials have shown the crucial importance of T-cell responses in limiting viral replication, including the control of virus transcription in latently infected cells that persist under ART [14,15,16] and partial protection against retroviral challenge [17,18].

Extensive efforts are being continued in order to develop an effective therapeutic vaccine against HIV-1 which would induce a lasting control of HIV-1 infection to partially or completely eliminate the need for lifelong ART [19,20,21,22,23]. Lately, a clinical study of the novel therapeutic HIV-1 vaccine consisting of the replication-defective HIV (HIVAX), given in the context of viral suppression under ART, provided evidence that therapeutic HIV vaccination in subjects under long-term antiviral suppression can reduce immune activation/chronic inflammation and latent infection (NCT01428596) [19]. Another recent study of a therapeutic HIV-1 DNA vaccine expressing SIV Env, Gag, and Pol delivered by intradermal electroporation with a novel combination of adjuvants and immunomodulators demonstrated the vaccine’s capacity to augment T-cell immunity in the blood and gut-associated lymphoid tissue and partially protect SIV-infected rhesus macaques against viral rebound after treatment interruption [24]. The lower viral load among controllers during analytical treatment interruption significantly correlated with higher levels of specific polyfunctional CD8+ T-cells expressing three or more effector functions in both blood and mesenteric lymph nodes. Altogether, these data indicate that immune responses, specifically mucosal, combined with an effective ART can play a crucial role in the viral containment post-ART, highlighting the need for therapeutic vaccines and adjuvants that would restore the functionality of peripheral and mucosal T-cell response in people living with HIV-1. Additional measures to control HIV-1 replication in HIV-infected patients are needed in view of the increasing resistance of HIV-1 to currently employed antiretroviral drugs, as well as drug-associated adverse reactions and toxicity.

Of note, the immune system of HIV-infected individuals recognizes DR mutations, which sometimes are even more immunogenic than the parental wild-type sequences [25,26,27]. A potent immune response against DR HIV-1 antigens can suppress viral replication and evolvement towards drug resistance, suggesting an option of therapeutic vaccination against existing and prophylactic vaccination against anticipated DR HIV-1 variants [28,29]. The main targets of these vaccines are HIV enzymes, reverse transcriptase, integrase, and protease.

HIV-1 protease (PR) plays a crucial role in the viral replication cycle as it cleaves Gag and Pol polyproteins into their mature forms, enabling the formation of infectious virions. The enzyme belongs to the pepsin-like aspartic proteases [30,31]. Its catalytic activity depends on protein dimerization; the active center is formed by aspartic acid residues (D25) of the two subunits. Protease inhibitors (PI) blocking the catalytic activity of PR and disrupting the viral replication cycle have been used as a part of the combined ART since the late 1990s [32]. With the development of non-nucleoside reverse transcriptase inhibitor (NNRTI) class drugs, and more recently, integrase inhibitors, PIs have been reserved mostly for the second- and third-line therapy combinations for patients failing the initial treatment regimen [33]. Alarmingly, the latest studies demonstrate an increase in the frequency of acquired DR mutations in PR resulting in the failure of these therapies, common in resource-limited settings [34,35]. Such a failure requests a switch to third-line ART which induces more side effects and requests more resources for the provision (as it costs eighteen times more than the lowest price of the first, and seven times more than the lowest price of second-line ART) [36,37,38,39]. These considerations specifically make the development of a therapeutic HIV-1 vaccine, which would prevent or hinder the evolvement of DR mutations in PR, imperative. 

In this study, we developed a candidate vaccine against DR HIV-1 based on the consensus sequence of PR of the HIV-1 clade A FSU_A strain, and evaluated its immunogenicity, immunotoxicity, and protective potential in mice by challenging immunized animals with syngenic PR-expressing tumor cells mimicking viral infection. Our data demonstrate that as a DNA immunogen, inactive DR HIV-1 PR is nontoxic and highly immunogenic for CD8+ T cells. The CD8+ T-cell response against the PR epitope containing the DR mutation provides close-to-sterilizing protection against tumor cells expressing the protein identical to that encoded by the plasmid immunogen, demonstrating that DNA immunization can induce an immune response recognizing single DR mutations. The study also demonstrates the applicability of syngenic tumor cells expressing HIV-1 proteins for testing the efficacy of HIV-1 vaccine candidates in mice.

## 2. Methods

### 2.1. Design and Generation of Consensus FSU-A Protease Genes

PR sequences from treatment-naïve patients from the FSU infected with the HIV-1 FSU-A strain were selected from HIV-1 databases (Los Alamos, Genbank). A set of sequences (n = 206) attained from Ukraine (UA), Russia (RU), Uzbekistan (UZ), Kazakhstan (KZ), Belarus (BL), Georgia (GE), Estonia (ES), Moldova (MD), and Azerbaijan (AZ) in 1997–2003 were compared to a set of sequences (n = 120) attained from KZ, KG, UZ, and RU in 2009–2015. Sequences within each of these subsets were aligned using MUltiple Sequence Comparison by Log-Expectation (MUSCLE; [40]; www.ebi.ac.uk/Tools/msa/muscle/ accessed on 18 May 2018), and consensus sequences were generated with Geneious v.8.1.2 (Auckland, New Zealand, www.geneious.com accessed on 18 May 2018).

A consensus gene optimized for PR_A expression in the human cells was synthesized by Evrogen (Moscow, Russia). For prokaryotic expression, a PR_A coding sequence was cloned into vector pET15b (Merck Millipore, Darmstadt, Germany) generating pET15bPR_A, and for eukaryotic expression, into expression vector pVAX1 (Life Technologies, Stockholm, Sweden) generating pVaxPR_A (Appendix A). In both schemes, insertions were performed after cleaving plasmid vectors with BamHI and EcoRI.

The primary drug resistance mutation pattern M46I/I54V/V82A common for FSU-A and conferring high levels of resistance to PR inhibitors used in 1st- and 2nd-line ART was selected based on the data available in the Stanford HIV drug resistance database (Stanford HIV Drug Resistance Database at http://hivdb.stanford.edu/ accessed on 18 May 2018). Mutations were introduced into the pVaxPR_A gene by site-directed mutagenesis using Agilent Technologies as combinations of two (M46I/I54V; “PR_A2mut”) or three mutations (M46I/I54V/V82A; “PR_A3mut”) (Appendix A). Next, we introduced PR inactivation D25N mutation which significantly reduces PR activity and increases the expression of PR in eukaryotic cells [30,41]. Mutagenesis generated variants PR_Ai2mut with D25N/M46I/I54V and PR_Ai3mut with D25N/M46I/I54V/V82A mutations. The presence of DNA inserts encoding PR_A variants was confirmed by sequencing. Active and inactivated proteases of the HIV-1 clade B HXB-2 strain were recloned from plasmids pKCMVPR_B and pKCMVPR_Bi into the pVAX1 vector to generate pVax_PR_B and pVax_PR_Bi, respectively [41] (Appendix A). Plasmids to be used for DNA immunization were purified using Endotoxin-free Megaprep kit (Qiagen, Darmstadt, Germany) according to the manufacturer protocol.

### 2.2. Preparation of Lentiviral Vectors Encoding PR Variants

Coding sequences for PR_A, PR_Ai, PR_A2mut, PR_Ai2mut, PR_A3mut, and PR_Ai3mut were recloned into the lentiviral vector pRRLSIN.cPPT.PGK (Addgene plasmid #12252; a gift from Dr Trono) generating pLVPR_A, pLVPR_Ai, pLVPR_A2mut, pLVPR_Ai2mut, pLVPR_A3mut, and pLVPR_Ai3mut, respectively. Lentiviral particles were produced by the transient transfection of HEK293T cells by Eurogen (Moscow, Russia) as described elsewhere [42] and concentrated 10-fold with Amicon Ultra-15 100K centrifuge concentrators (Merck-Millipore, Darmstadt, Germany). Infectious viral particles were formed only by pLVPR_A3i; their titer was determined on HT1080 cells with quantitative real-time PCR [42] using standard samples of HT-1080 DNA with a known number of viral genome copies. 

### 2.3. Recombinant HIV-1 Proteases and Protease-Specific Antibodies

Variants of PR of HIV-1 FSU_A with and without inactivation and DR mutations were obtained through expression of the respective plasmids in *E. coli*. For this, competent *E. coli* cells of the Rosetta (DE3) strain (Novagen, Darmstadt, Germany) were transformed by heat shock with respective pET15b-based plasmids (Appendix A) [43]. A single colony of transformed bacterial cells was inoculated into Luria broth (LB) medium with ampicillin (amp, 100 µg/mL) and incubated overnight at 37 °C and 200 rpm. The overnight culture was diluted into fresh LB/amp medium 1:10 and cultured as described until reaching an optical density of 0.6 to 1.0 at 600 nm. Protein synthesis was induced by culture with IPTG for 3 h. Bacterial pellets were sequentially resuspended in lysis buffers A (50 mM Tris HCl pH 8.0, 0.2 M NaCl, 2% Triton X-100, 1 mM EDTA, 1 mM PMSF), B (50 mM Tris HCl pH 8.0, 0.2 M NaCl, 2M Urea, 5 mM 2-mercaptoethanol) and C (50 mM Tris HCl pH 8.0, 0.5 M NaCl, 8M Urea, 5 mM 2-mercaptoethanol). In each buffer, cells were sonicated 4 times for 45 s with a pulse of 7 ms on ice using a Bandelin Sono Plus apparatus (Bandelin, Berlin, Germany) and then centrifuged for 30 min at 15,000× *g* at 4 °C. The His-tagged proteases from the inclusion bodies were purified by affinity chromatography using the Ni-NTA Purification System (Invitrogen) and buffer I (50 mM Tris HCl pH 8.0, 0.5 M NaCl, 8M Urea, 25 mM imidazole, 5 mM 2-mercaptoethanol), buffer II (50 mM Tris HCl pH 8.0, 0.5 M NaCl, 8M Urea, 10 mM imidazole, 5 mM 2-mercaptoethanol), and elution buffer (50 mM Tris HCl pH 6.8, 0.25 M NaCl, 6M Urea, 0.5 M imidazole, 1 M EDTA, 5mM 2-mercaptoethanol). The resulting proteins were dialyzed against buffer 1 (20 mM Tris HCl pH 6.8, 0.2 M NaCl, 2.5 M Urea, 0.1 M imidazole, 20 mM EDTA, 5mM 2-mercaptoethanol) for 1 h, then buffer 2 (20 mM Tris HCl pH 6.8, 0.5 M NaCl, 25mM EDTA, 5 mM 2-mercaptoethanol) overnight, and then storage buffer (20 mM Tris HCl pH 6.8, 0.15 M NaCl, 20 mM EDTA, 5mM 2-mercaptoethanol) for 30 min, then aliquoted and stored at −80 °C. The recombinant PR of the HIV-1 clade B HXB-2 strain was kindly donated by the National Institute for Biological Standards and Control (NIBSC) and the Programme EVA Centre for AIDS Reagents (Hertfordshire, UK) (https://www.hivreagentprogram.org/Catalog/HRPProteins/ARP-11781.aspx accessed on 10 March 2013).

Antibodies against PR_Ai were raised in-house following the ethical routine and experimental protocol described by us earlier [44,45]. PR_Ai was used to immunize two rabbits (#89, #90). All immunizations were performed with freshly prepared protein preparations due to protein instability. Aliquoted sera were stored at −20 °C until further use. Antibodies against the PR of HIV-1 clade B were purchased: rabbit polyclonal sera, from Fitzgerald, GA, USA, No 20-000801, and monoclonal antibody, from Exbio (Vestec, Czech Republic), clone 1696.

### 2.4. Peptides Representing PR Epitopes and In Silico Assessment of the Effects of DR Mutations on Immunogenicity

The consensus PR sequence was aligned with sequences of the known B- and T-cell epitopes in HIV-1 PR using the epitope data from Los Alamos HIV (www.hiv.lanl.gov accessed on 7 March 2013) and IEDB databases (http://www.immuneepitope.org accessed on 10 March 2013). Peptides homologous to the known epitopes and predicted by the IEDB Immunogenicity prediction tool (http://www.iedb.org accessed on 10 March 2013) to be recognized in mice were selected and represented by synthetic peptides (GL Biochem Ltd., Shanghai, China) (Figure 1). Using the IEDB Immunogenicity prediction tool, we generated prediction scores for all 9-mer peptides spanning the regions of M46I, I54V, and V82A mutations. The first peptide spanning the region had the mutated amino acid residue on the C- and the last, on the N-terminus (Appendix A). Four peptides with the highest scores were selected, and their scores were summed into an integral immunogenicity score (IIS). IIS values were then compared for the variants with and without DR mutation for each of the three regions. 

### 2.5. Eukaryotic Cell Transfection

HeLa cells were transfected using the Lipofectamine LTX reagent (Life Technologies) and cultured for 24 h prior to the addition of proteasomal inhibitor MG132 (5 μM, Sigma, Stockholm, Sweden) for an additional 18 h, or simply uninhibited for a total of 48 h, prior to harvest and quantification. Cells were lysed by heating (10 min at 90 °C) in Tris-glycine sample buffer (Life Technologies, Stockholm, Sweden) supplemented with 5% β-mercapto-ethanol (Sigma–Aldrich, Stockholm, Sweden) and analyzed through Western blotting.

### 2.6. Analysis of Protease Expression through Western Blotting

To detect PR expression in *E. coli* and eukaryotic cell lysates, the protein samples were loaded on a 1.0 mm thick 18% Tris-glycine gel and separated by electrophoresis in Tris-glycine running buffer (both by Invitrogen). The proteins were subsequently blotted onto a PVDF membrane in Tris-glycine transfer buffer (Invitrogen) and the membrane was blocked in 2.5% dry skimmed milk suspended in the phosphate-buffered saline containing 0.1% Tween 20 (Sigma-Aldrich Stockholm, Sweden; PBS-T). Blots were first incubated with pooled hyperimmune sera of rabbits № 89 and 90 (described above; 1:4000), or commercial anti-PR 20-000801 (Fitzgerald; 1:500) followed by the secondary horseradish peroxidase (HRP)-conjugated anti-rabbit antibody (Ab6721, Abcam, Cambridge, UK). Alternatively, staining was carried out with mouse monoclonal IgG1 antibody clone 1696 (Exbio; cat 11-302; 1:2000) followed by the secondary HRP-conjugated anti-mouse antibody (Ab6789, Abcam). In between incubations, the blots were washed 3 times in PBS-T. The membranes were developed using the ECL Plus Western blotting detection system (GE Healthcare, Chicago, IL, USA). To normalize the signal, the membranes were stripped with buffer containing 62.5 mM Tris–HCl, 2% sodium dodecyl sulfate and 100 mM β-mercaptoethanol, and restained with a mouse monoclonal anti-actin antibody (AC-15, Sigma-Aldrich, Stockholm, Sweden) and a secondary HRP-conjugated anti-mouse antibody Ab6789 as described above. Protein bands were quantified using the Image J software (NIH, Bethesda, MD, USA).

### 2.7. Evaluation of Protease Activity

Forty-eight hours post-transfection with pVax-based vectors encoding PR_A variants, HeLa cells were lysed using a cell extraction kit under non-denaturing conditions (Biovision, Milpitas, CA, USA). The protease activity of the transfected cells was determined by a FRET assay using the SensoLyte 490 HIV-1 protease assay kit containing fluorescent substrate (Anaspec, Fremont, CA, USA). To measure the activity of recombinant PR_A protein variants, they were resuspended in the assay buffer. Substrate cleavage was monitored fluorimetrically with excitation at 340 nm and emission at 490 nm using an M1000 Pro plate reader (Tecan, Maennedorf, Switzerland).

### 2.8. Lentiviral Transduction of 4T1luc2 Cells and Isolation of PR_A-Expressing Clones

Viable lentiviral particles could be generated only for the PR_Ai3mut variant. Lentiviral constructs encoding other PR_A variants did not yield viable lentiviral particles. Lentiviral particles encoding for PR_A3i were used to transduce 4T1luc2 cells at a multiplicity of infection of 20. A lentivirus preparation encoding PR_Ai3mut was used to transduce murine mammary gland adenocarcinoma cells expressing firefly luciferase 4T1luc2 (“Bioware Ultra Cell Line 4T1luc2,” Caliper, Hopkinton, MA, USA; http://www.caliperls.com/assets/014/7158.pdf accessed on 20 February 2015) with a multiplicity of infection (MOI) of 20 transducing units per cell. The transduced cells were cloned to single cells by limiting dilution in 96-well plates generating monoclonal populations of 4T1luc2 subclones. The presence of PR_Ai3mut-encoding sequence in the DNA of the resulting subclones of 4T1luc2 was confirmed by PCR with a pair of primers specific for the lentivector backbone and flanking the PR insert. Subclones were cultured in the full RPMI-1640 medium with 10% FBS and 100 mg/mL penicillin/streptomycin mix at 37 °C in 5% CO_2_ and split every 2–3 days. One of the subclones, 4T1luc2_PR_Ai3mut_20.2 (dubbed PR20.2 and containing a genomic insert of PR_Ai3mut DNA), was selected for further experiments. 

### 2.9. Experiments in Laboratory Mice

Experiments in laboratory mice were carried out in compliance with the bioethical principles adopted by the European Convention for the Protection of Vertebrate Animals Used for Experimental and Other Scientific Purposes (Strasbourg, 1986). Experimental procedures were approved by the Commission for Animal Research of Northern Stockholm (Stockholms Norra Djurförsöksetiska Nämnd, permit N66_13 dated 16 May 2013). In series I, II, and IV (Table 1), BALB/c (H2-Dd) mice (females, 8 weeks old) were purchased from Charles River Laboratories (Sandhofer, Germany) and housed at the Astrid Fagrius Laboratory (Karolinska Institute, Stockholm, Sweden). In series III assessing immunotoxicity (Table 1), BALB/c mice were bred and housed at the Scientific Center of Biomedical Technologies of the Federal Medical and Biological Agency of Russia (https://fmbafmbc.ru/en/scientific-activities/research-unit/the-center-for-biomedical-technology/ accessed on 15 January 2018). Mice were housed in environment-enriched cages with 5–8 animals per cage under a 12 h/12 h light–dark cycle with ad libitum access to water and food. All procedures were evaluated as having a low-to-average degree of difficulty. Mice were regularly controlled for skin and fur changes, macroscopic alterations at the site of immunization, as well as food and water intake and weight development. Possible mouse discomfort under immunization, monitoring, and sample collection was relieved in series I, II, and IV by inhalation anesthesia with a 1L/min flow of oxygen containing 4% isoflurane at the induction and 2.5% at the maintenance stage, administered through the facial masks, and in series III by injections of Zoletil 100 at 40 mg/kg and Domitor at 35 mcg/kg. Animals were sacrificed by cervical dislocation.

### 2.10. DNA Immunization of Mice 

Groups of 8-week-old BALB/c mice were immunized with plasmid immunogens, empty vector (20 µg in 20 µL PBS per injection), or PBS as depicted in Table 1. Solutions were delivered intradermally using a 29G syringe (Micro-Fine U-100 insulin; BD, Franklin Lakes, NJ, USA). Mice in series I, II, and IV received injections at two sites to the left and to the right of the back of the tail, and in series III at three sites: two by the back of the tail and one in the right shoulder. All injections were followed by electroporation carried out as described by us earlier [45]. Pulses were delivered using DERMAVAX (Cellectis, Paris, France) or a CUY21EditII electroporator with BEX fork-plate electrode (BEX Ltd., Tokyo, Japan). At the experimental endpoint, the mice were bled through the tail vein; the blood samples were collected into Microtainer gel tubes (BD, Franklin Lakes, NJ, USA). After bleeding, the mice were sacrificed, the spleens were excised, and splenocyte cultures were prepared as described by us earlier [45].

## 3. Humoral Immune Responses Assessed by ELISA

Microtainer gel tubes were subjected to centrifugation for 5 min at 2900× *g* to separate sera. The sera were diluted serially ranging from 1/200 to 1/50,000 in a Scan buffer (PBS containing 0.5% BSA, 2% goat serum, 0.05% Tween 20) and transferred onto Nunc Immuno 96 MicroWell Solid Plates (Thermo Scientific, Waltham, MA, USA) coated with recombinant protease variant (0.2 µg/mL in carbonate/bicarbonate buffer pH9.3 for 16–20 h at 6 °C). The loaded plates were incubated at 7 °C overnight, then washed with PBS containing 0.05% Tween 20 (Wash buffer). Mouse serum reactivity with PR was detected using secondary polyclonal goat anti-mouse IgG HRP conjugate (Dako, Glostrup, Denmark) diluted 1:2000 in Scan buffer. After 1 h of incubation at 37 °C, the plates were washed 5 times and treated with 100 µL per well of 3, 3′, 5, 5′-Tetramethylbenzidine (TMB) substrate (Dako) until color was sufficiently developed (max 15 min at room temperature). The reaction was stopped by 50 µL of 2.5 M sulfuric acid. The optical density was assessed spectrophotometrically at a dual wavelength of 450 and 650 nm. A cut-off for positivity was defined as an average signal (signal = optical density at 450–650 nm; in OD) demonstrated by the sera of the vector-immunized mice.

## 4. Assessment of T-Cell Immune Response by Fluorospot and Flow Cytometry

*Fluorospot* Splenocytes were tested for the capacity to produce IFN-γ and IL-2 in response to stimulation with PR-derived synthetic peptides (Figure 1) at a concentration of 10 μg/mL using the dual IFN-γ/IL-2 Fluorospot test (Mabtech, Nacka Strand, Sweden). The tests were performed as described by the manufacturer. Concanavalin A (ConA; Sigma; 5 µg/mL) and the medium alone were used as positive and negative controls. Cytokine-producing spot-forming cells (SFC) were registered and counted using the AID ELISpot reader system (Autoimmun Diagnostika GmbH, Strassberg, Germany). An overlay of images was created to detect and quantify dual IFN-γ/IL-2 cytokine-secreting cells [46]. The number of SFCs per 10^6^ splenocytes detected in response to stimulation with the PR peptide and control antigens (SFC/10^6^) was calculated for each sample. The specific signal of splenocytes of a mouse in response to the PR peptide was calculated by subtracting from the above value the SFC/10^6^ detected in response to the stimulation of these splenocytes with the medium alone. 

*Flow cytometry* All the reagents used in flow cytometry with intracellular cytokine staining (ICCS) were from BD Biosciences (Franklin Lakes, NJ, USA) if not mentioned otherwise. The splenocytes were purified and frozen at −80 °C. A week later, the cells were thawed and subjected to stimulation with PR-derived peptides by incubation for 4 h with an equimolar mixture of peptides representing mouse CD4+ and CD8+ epitopes 10 ug/mL each (Figure 1), or positive controls, phorbol myristate acetate (PMA, Sigma, Darmstadt, Germany) at 50 ng/mL and Ionomicin (Sigma) at 1 µg/mL, or the medium alone. The medium with stimuli was supplemented with GolgiPlug (Becton Dickinson) containing Brefeldin A (1:1000). The stimuli were diluted in complete culture media consisting of RPMI supplemented with 5% FBS, 100 U/mL penicillin, 100 μg/mL streptomycin, and 0.3 mg/mL glutamine (Gibco, ThermoFisher, Waltham, MA, USA). Ten minutes before the end of incubation, anti-mouse CD16/CD32 antibody (#553142) was added to block non-antigen-specific binding of immunoglobulins to Fcγ receptors. Before proceeding to staining the surface molecules, the cells were stained for viability using the Fixable Viability Stain 660 (FSV660, cat. #564405) as recommended by the manufacturer. Surface staining was performed by incubating the stimulated cells with FITC-conjugated anti-mouse CD8a (#553031), APC-H7-conjugated anti-mouse CD4 (#560181), and PerCP-conjugated anti-mouse CD3 (#553067) antibodies. After that, the cells were fixed and permeabilized at room temperature for 20 min in 100 μL of Cytofix/Cytoperm solution, washed with Perm/Wash buffer (#554723), and stained at 4 °C for 30 min with PE-conjugated anti-IFN-γ (#554412), BV421-conjugated anti-IL-2 (#562969), and BV510-conjugated anti-TNF-α (#563386) antibodies specific to murine cytokines. The samples were then analyzed on a FACSVerse cytometer (BD Biosciences, Franklin Lakes, NJ, USA) using FACSDiva software. The data were exported as FCS3.0 files and analyzed in openCyto [47]. A general lymphocyte area was defined and single living cells within this population were defined by the lack of FSV660 staining. From the viable population, single cells were defined and that population was further gated according to the expression of surface markers, such as CD3, CD4, and CD8 and the production of cytokines, such as IFN-γ, IL-2, and TNF-α. The frequencies of CD8+ and CD4+ cells producing cytokines in response to PR-specific stimulation were quantified, and the values for unstimulated cells were subtracted.

### 4.1. Immunotoxicity Tests

Groups of 20 to 22 g BALB/c mice (n = 12) were immunized as described in series III (Table 1). The general health condition of the animals was assessed by a veterinary doctor daily. An in-depth clinical examination with scoring was performed weekly according to the parameters shown in Appendix A. Body mass was measured daily until day 7 and weekly afterwards. On day 1 after the last immunization, half of the animals (n = 6), and on day 13 after the last immunization, the other half (n = 6), were euthanized with subsequent testing of clinical and biochemical parameters of blood, an assessment of bone marrow composition, and measurement of the mass of lymph nodes, thymus, and liver. Blood samples were obtained by decapitation on days 1 and 12 after booster immunization following 18 h of starvation. A complete blood count, including red blood cells (RBC), hematocrit (HCT), platelets (PLT), hemoglobin (HGB), white blood cells (WBC), lymphocytes (LYM), unclassified leukocytes (MID), and granulocytes (GRAN), was performed using a Medonic CA-620 hematology analyzer (Boule Medical, Sweden) according to the manufacturer’s protocols. Bone marrow smears were stained using the Romanowsky–Giemsa procedure and cell composition was assessed using microscopy [48]. Biochemical analysis was performed, including measurement of the levels of alanine aminotransferase (ALT), aspartate aminotransferase (AST), alkaline phosphatase (ALP), lactate dehydrogenase (LDH), total protein (TP), albumin (ALB), urea (UREA), glucose (GLC), cholesterol (CHOL), triglycerides (TG), sodium (Na), and potassium (K). The analysis was performed using a Stat Fax 4500+ biochemical analyzer (Awareness technology, Ramsey, MN, USA) according to the manufacturer’s protocols.

### 4.2. Tumor Implantation and Follow-Up of Tumor Growth in Naïve and Immunized Mice

Groups of DNA-immunized and control BALB/c mice were implanted with murine mammary gland adenocarcinoma cells 4T1luc2 stably expressing PR_Ai3mut, subclone PR20.2 (Table 1, series IV). The implantations were performed 1.5 weeks post-booster-immunization. Prior to implantation, PR20.2 cells were resuspended in RPMI-1640 at a concentration of 2 × 10^5^ cells/mL; 50 μL of the suspension (containing 10^5^ cells) was subcutaneously injected into the BALB/c mice at two sites to the right and to the left of the base of the tail, at least 1 cm over the site of DNA immunization. Tumor growth was assessed by regular bioluminescent imaging (BLI) on IVIS Spectrum (Perkin Elmer, Waltham, MA, USA) following the protocols described by us earlier [49,50]. For the quantification of photon emission, a frame was created of the size sufficient to enclose all photon emissions from the mice throughout the entire course of the study (region of interest, ROI); this ROI frame was applied to all images. The total flux from ROI was quantified using Living Image 3.2 software (Caliper Life Sciences, Waltham, MA, USA). An average of the two immunization sites per mouse was calculated. Tumor size was assessed by morphometric measurements carried out at regular intervals using calipers; tumor volume was calculated using the standard formula for xenograft volume, V = (xy^2)/2. Mice were weighed at each monitoring point. 

At the experimental endpoint, mice were anaesthetized as described above. A freshly prepared solution of XenoLight D-luciferin potassium salt (Perkin Elmer) in PBS was injected into the mice intraperitoneally in an amount of 150 mg/kg. Ten minutes later, the mice were euthanized by cervical dislocation by a skilled animal handler. The fur and skin of the mice were disinfected with 70% ethanol. After that, the tumors, spleen, liver, and lungs known to be affected by the distant metastasis in the 4T1 tumor model [51,52] were dissected with surgical scissors. The tumors and organs were transferred into the wells of a 24-well tissue culture test plate (Wallac, Turku, Finland) containing 2 mL of RPMI-1640 medium. The ex vivo bioluminescent imaging of the tumors and organs was performed as described for in vivo imaging. Thereafter, the tumors and all organs, except for the spleens, were transferred to 5 mL of 4% formaldehyde solution in PBS, incubated for 48 h at +6 °C, then washed five times with PBS and paraffin-embedded. The spleens were washed from luciferin with PBS, transferred into Petri dishes containing 3 mL of RPMI and ground with a syringe plunger; single-cell splenocyte cultures were prepared for T-cell tests.

### 4.3. Tumor Histology and Histological Assessment of the Metastases

FFPE blocks were prepared from the formalin-fixed tumor tissues and murine lungs and livers and sectioned on a cryostat microtome according to the standard protocols (https://www.protocolsonline.com/histology/sample-preparation/paraffin-processing-of-tissue/ accessed on 5 August 2016). Sections mounted on slides were dewaxed, rehydrated, and stained with Mayer’s hematoxylin solution, then washed, rinsed, and counterstained with eosin Y solution; after that, they were dehydrated, washed with absolute alcohol, and covered with cover slips for microscopic evaluation. Histological evaluation was based on the standard parameters such as acinar formation, nuclear size, and pleomorphism and mitotic activity. The grade of the tumors was calculated according to [53]. The slides were examined using light microscopy (Leica DM500, Wetzlar, Germany). Formalin-fixed, paraplast-embedded liver tissues were used to diagnose and evaluate the formation of metastases. For each mouse, the area of tumor metastases was quantified in 25 high-power (400×) microscope fields of hematoxylin–eosin-stained slides by computer-assisted morphometry using specialized NIS-Elements software (Nikon, Tokyo, Japan). Total DNA was isolated from five freshly prepared sections using Allprep DNA/RNA FFPE kit (Qiagen, Hilden, Germany). The presence of the sequence encoding PR_A variants was confirmed by PCR using a pair of primers, one specific for the lentivector backbone and the other to the PR-coding sequence.

### 4.4. Statistical Analysis

Data were presented as individual entries, expressed as median values, or as mean ± standard deviation (SD). Graphical presentation of the data and statistical calculations were performed using Microsoft Excel, GraphPad Prism version 9.4.0 software (GraphPad Software, San Diego, CA, USA), STATISTICA AXA 10.0 software (StatSoft Inc., Tulsa, OK, USA), and Statistical Package for Social Sciences (IBM SPSS, version 17.0, Armonk, NY, USA). Continuous but not normally distributed variables, such as the number of cytokine-producing spot-forming cells, total flux, and different parameters of clinical and biochemical blood analysis, bone marrow composition, and organ mass were compared using the nonparametric F-test and Kruskal–Wallis and Mann–Whitney U-tests, with Bonferroni or Holm and Hochberg corrections applied for multiple comparisons or by two-way ANOVA with Dunnett’s or Tukey’s multiple comparisons corrections. Linear correlations between variables were analyzed using the Spearman rank-order test, if not stated differently. A statistical significance threshold of 0.05 was used throughout the study.

## 5. Results

### 5.1. Design of the Consensus Protease of HIV-1 FSU_A Strain and Its Variants with Drug-Resistance-Conferring Mutations

Consensus sequences of the protease of the HIV-1 FSU_A strain (PR_A) were generated based on two sets of PR sequences, one collected in the territory of the former Soviet Union and deposited in the HIV Los Alamos Database and Genbank in 1997–2003 (first; n = 206), and the other in 2009–2015 (second; n = 120) (Figure 1A,B). Two sets generated one and the same consensus sequence (Figure 1C). A synthetic gene encoding enzymatically active consensus PR_A expression optimized for mammalian cells was designed, synthesized, and cloned into the pET15b vector for prokaryotic, and into pVAX1 for eukaryotic, expression, generating pET15bPR_A and pVaxPR_A, respectively (Appendix A).

Next, we selected primary resistance mutations to protease inhibitors in use in the territories of FSU_A circulation [54] (Stanford HIV Drug Resistance Database. at http://hivdb.stanford.edu/ accessed on 18 May 2018). Common mutations were selected rendering high levels of resistance to these drugs: M46I, I54V, and V82A, in the combinations M46I/I54V and M46I/I54V/V82A. Plasmids pET15b-PR_A and pVax-PR_A were subjected to site-directed mutagenesis introducing M46I/I54V (PR_A2mut) or M46I/I54V/V82A (PR_A3mut) (Appendix A). To increase the level of PR_A expression, we subjected PR genes to the second round of site-directed mutagenesis to introduce mutation D25N abrogating PR activity [30], as it was earlier shown to lead to up to a 100-fold increase in the levels of PR expression in eukaryotic cells [41]. Mutagenesis generated plasmids encoding the inactive PR variants PR_Ai, PR_Ai2mut, and PR_Ai3mut (Figure 1C; Appendix A). 

To assess the immunogenicity of the PR_A variants, we selected synthetic peptides representing known B- and T-cell epitopes of PR predicted to be recognized in mice (Immune Epitope Database at http://www.iedb.org/ accessed on 18 May 2018) with and without drug resistance (DR) mutations (Figure 1C). An in silico assessment of the immunogenicity of PR regions harboring DR mutations was made to predict the impact of DR mutations on the outcome of immunogenicity. For this, we introduced the integral immunogenicity scores (IIS) for each of the epitopic regions using the IEDB Immunogenicity prediction tool (http://www.iedb.org accessed on 18 May 2018) (Appendix A). The IIS of the region encompassing V82A was unaffected by the mutation. Mutation M46I was predicted to increase and I54V to decrease the IISs of the respective regions (Appendix A). Overall, the analysis predicted that DR mutations would not exert a unidirectional change towards either higher or lower PR immunogenicity.

### 5.2. Synthetic Genes’ Direct Expression of Enzymatically Active HIV-1 Proteases in Bacteria

Our first step was to characterize the synthetic consensus PR_A and its mutant variants as enzymes. To this end, we expressed PR_A in *E. coli* transformed with pET15bPR_A which encodes PR_A (Appendix A). The recombinant protein with an expected molecular mass of 11 kDa was recognized by the polyclonal antibodies directed against PR_B, albeit with low sensitivity (Figure 2A), but not with the monoclonal antibodies against PR_B (clone 1696; Figure 2B). Low or no cross-reactivity of anti-PR_B antibodies with PR_A could be due to amino acid differences between PR_A and PR_B in the region harboring dominant protease epitopes at aa 1–13 and 36–46 [55] (Figure 1C). For further analysis of protein expression, we raised PR_A-specific antibodies in rabbits. The hyperimmune polyclonal rabbit sera detected as low as 2.5 ng of recombinant PR_A (Figure 2C, panels 1–4), and both enzymatically active and inactivated DR variants of PR_A (Figure 2D), but poorly recognized the PR of the HIV-1 subtype B isolate HXB2 (kindly donated by the NIBSC HIV-1 reagent collection) (PR_B; Figure 2D).

We further assessed the enzymatic activity of the PR_A variants using an HIV-1 protease activity fluorescence resonance energy transfer (FRET) assay (SensoLyte 490, Anaspec, Fremont, CA, USA). The assay demonstrated that all consensus PR_A proteins were enzymatically active, although their activity was decreased by 20–40% compared to that of PR_B (Figure 2E). We attributed the latter to the incomplete refolding and/or re-aggregation of PR_A variants in the process of purification; these factors were earlier shown to result in decreased PR activity [56]. Earlier studies demonstrated that a single V82A mutation causes up to a two-fold decrease in enzyme activity ([57] and references therein); however, mutations at aa positions 46 and 54 compensate for the loss [58], which could explain the similar activity of all three PR_A variants. To conclude, the synthetic PR_A-genes encoded proteins of expected molecular mass that had enzymatic activity characteristic to HIV-1 protease and were stained with PR-specific antibodies.

### 5.3. Suppression of Protease Activity Increases Protease Content in the Expressing Cells

In the next step, coding sequences for PR_A variants were recloned into the DNA vaccine vector pVAX1 and assessed for expression in eukaryotic cells. HEK293 cells transfected with plasmids encoding PR_A variants expressed the proteins specifically stained with antibodies against recombinant PR_A (Figure 3A). The cellular content of all active PR variants was very low (Figure 3A). We inactivated all PR_A variants by the introduction of the inactivating mutation D25N, earlier shown to increase the expression of PR_B [41]. For PR_A as well, the introduction of D25N reduced the protease activity by up to 70% (Appendix A), and significantly increased the expression of all three variants (Figure 3C). The strongest effect was exerted on PR_A3mut; the intracellular level of its inactivated variant increased 15–20 times compared to all other PR_A variants, while the intracellular levels of PR_Ai and PR_A2imut were still quite low (Figure 3A,C).

The inhibition of proteasomal processing prevents the degradation of viral proteins and increases the efficacy of HIV-1 infection, including the release and maturation of HIV-1 gag polyprotein through its effect on HIV-1 protease [59] unrelated to PR activity [60]. We tested if HIV-1 PRs could be stabilized by treatment with an inhibitor of proteasomal-processing, MG132. The treatment of cells expressing PR gene variants with MG132 led to a significant increase in the level of expression of all protease variants, except for PR_Ai3mut and PR_Bi initially expressed at a high level. The latter indicated their diminished susceptibility to proteasomal cleavage compared to other PR variants (Figure 3B,C).

Thus, all PR_A genes directed the expression of the encoded protein variants in eukaryotic cells, albeit at varying levels, related to the efficacy of autocleavage and susceptibility to degradation by the proteasome. Low levels were related to proteolytic degradation. The introduction of mutation D25N inhibiting the autocleavage and intervening with the proteasomal cleavage of the triple mutant resulted in a significant increase in the protein levels of mutants in the expressing cells.

### 5.4. Immunogenic Performance of Plasmids Encoding PR_A Variants in Mice

The performance of the synthetic PR_A genes in DNA immunization was assessed in BALB/c mice in two independent series of immunizations (series I and II, Table 1). All plasmids were delivered by intradermal (id) injections followed by electroporation (EP).

#### 5.4.1. PR_A and PR_Ai Variants Are Strongly Immunogenic after Single DNA Immunization

To assess HIV-1-specific immune responses induced by immunization with the parental PR_A in comparison with the PR of subtype B (PR_B), female BALB/c mice in groups of five were immunized by id injections of either PR_A or its variant with the inactivation mutation D25N (PR_Ai), or with PR_B or its variant with the inactivation mutation (PR_Bi), or empty vector (series I, Table 1). The indirect ELISA with recombinant PR variants performed with sera collected at the experimental endpoint demonstrated that DNA immunization with PR_A variants induced no detectable antibody response. The specific cellular immune response was assessed by FluoroSpot, evaluating IFN-γ, IL-2, and dual IFN-γ/IL-2 secretion by splenocytes upon in vitro stimulation with PR-derived peptides or control antigens (Figure 1C and Figure 4). All PR gene variants except for PR_B induced IFN-γ responses to PR-derived peptides significantly higher than those observed in mice receiving the empty vector (2-way ANOVA; *p* < 0.0001) (Figure 4A–C). The same was observed for the frequencies of IL-2-secreting and dual IFN-γ/IL-2-secreting cells; all PR genes except for PR_B induced significantly higher responses than the empty vector (two-way ANOVA; *p* < 0.0001, Figure 4A–C). T-cell epitopes were localized in the regions aa 1–15 and 71–90 of PR, while no reactivity was detected against peptides presenting aa residues 31 to 70 (Figure 4A).

PR_A and PR_Bi induced a comparable, and PR_Ai a significantly higher, level of cytokine secretion upon stimulation with peptides common for two clades (two-way ANOVA; *p* < 0.0001, Figure 4B). Furthermore, the immune response was cross-reactive with respect to HIV-1 clades. No difference was observed between PR-immunized groups in the levels of cytokine secretion in response to stimulation with peptides specific to the PR of clade A and B (peptides A1–15 and B1–15, *p* > 0.05 in two-way ANOVA; Figure 4C). Additionally, stimulation with the clade-A-specific peptide A76–90 induced a strong cytokine response in mice immunized with both PR_A and PR_Bi variants (*p* > 0.05; Figure 4C). At the same time, IFN-γ, IL-2, as well as dual IFN-γ and IL-2 secretion in response to stimulation with peptide variants containing the DR mutation V82A (A71–85dr and A76–90dr) were significantly lower than that of nonmutated peptides (all *p* values <0.05), mostly falling below the cut-off level of 50 SFC/mln cells (Figure 4D). Thus, aa residues 46 and 54 lay outside, and aa residue 82 within, T-cell epitopes recognized in mice, but neither PR_A nor PR_Ai could induce an immune response able to recognize V82A mutation (Figure 4C).

#### 5.4.2. Homologous Prime–Boost Immunization with PR_A Gene Variants Induces a Cellular Response against Peptides Bearing Drug Resistance Mutations

Next, we assessed the immunogenicity of PR_Ai and its DR variants in BALB/c mice immunized following an optimized homologous prime/boost regimen [45]. Groups of BALB/c mice (n = 9) received PR_Ai, PR_Ai2mut (M46I/I54V), PR_Ai3mut (M46I/I54V/V82A), or an empty vector (series II, Table 1). The immune response of splenocytes stimulated with PR-derived peptides (Figure 1C) was assessed by a dual IFN-γ/IL-2 Fluorospot after prime and boost, and by multiparametric flow cytometry with intracellular staining (ICCS) for IFN-γ, IL-2, and TNF-α after the boost. Peptide A1–15 representing PR_A was recognized by the recipients of all PR_A variants, although in PR_Ai3mut-immunized mice, the response was weaker than in the other groups (Figure 5A–D). Mice immunized with PR_Ai and PR_Ai2mut also recognized nonmutated peptides AB71–85 and AB75–84 (no difference, *p* > 0.05; Figure 5A,C). No cytokine secretion was registered against peptides covering aa 31–56 with or without DR mutations either after prime (Figure 5A) or boost (Figure 5B). Although the DNA boost induced a weak response against A31–56dr in mice receiving PR_Ai2mut, it did not reach the level of significance (two-way ANOVA *p* > 0.05; Figure 5B). For peptides with the V82A mutation, AB71–85dr was recognized only in mice primed with the DR PR variants, and A76–90dr, only in mice primed with PR_Ai3mut (no difference between the plasmids, *p* > 0.05; Figure 5A,C). The boost with both DR PR variants enhanced the immune recognition of epitope(s) localized at aa 76–90 (two-way ANOVA *p* < 0.05; Figure 5D,E). The PR_Ai3mut boost specifically enhanced the IFN-γ response against A76–90dr (two-way ANOVA *p* < 0.05; Figure 5E, panel I). The IL-2 and dual IFN-γ/IL-2 response to A76–90dr after the boost remained unchanged, and the IFN-γ/IL-2 response against nonmutated AB75–84 decreased (*p* < 0.05; Figure 5E, panels II, III). Both observations suggested that mice immunized with PR_Ai3mut gradually developed an immune response against the V82A mutation.

We also characterized the IFN-γ, IL-2, and TNF-α cytokine response of the splenocytes of mice DNA-immunized with PR by flow cytometry with intracellular cytokine staining (Figure 6A). PR_Ai and PR_Ai2mut exhibited similar immunogenicity, as demonstrated by mono- and triple-IFN-γ/IL-2/TNF-α cytokine secretion by the CD8+ T cells of immunized mice in response to stimulation with peptides representing immunodominant T-cell epitopes of PR without DR mutations (A1–15, AB76–84, and A76–90; Figure 6A,B,E,F; Appendix A, Appendix A). PR_Ai2mut-immunized mice also exhibited significantly higher frequency of IFN-γ/IL-2/TNF-α-secreting CD8+ T-cells specific to peptide AB71–85, but not to its mutated variant AB71–85dr (Figure 6C,D, Appendix A, Appendix A). The immunogenicity profile of PR_Ai3mut-immunized mice was different. Their CD8+ T cells did not recognize peptides A1–15, AB71–85, AB76–84, and A76–90, but instead recognized peptides with the V82A mutation (Figure 6A–G, Appendix A, Appendix A). No specific cytokine secretion by the CD8+ T cells was detected in response to stimulation with peptides A31–56, A31–56dr, and A56–70 (Appendix A). The cytokine response to PR peptides by the CD4+ T cells was low-to-undetectable (Appendix A).

From these data, we concluded that (i) the introduction of the inactivation mutation D25N increased the immunogenicity of PR_A as was shown before for PR_B [41]; (ii) all the inactivated PR_A variants, both with and without DR mutations, were immunogenic on the cellular level, mostly for CD8+ T cells; (iii) DNA immunization with DR PR_Ai variants induced a cellular immune response recognizing peptides bearing the V82A mutation, not recognized by mice DNA-immunized with the parental PR_A; (iv) PR-encoding plasmids induced no antibody response.

#### 5.4.3. Protease A Variants Exert No Toxic Reactions/No Immunotoxicity in DNA-Immunized Mice

Next, we assessed the immunotoxicity of PR_Ai variants as DNA immunogens. For this, we assessed the effect of introducing plasmids by electroporation on the general health condition, body mass, parameters of complete blood count, biochemical blood parameters, bone marrow composition, and mass of immunocompetent organs in BALB/c mice primed and boosted with pVaxPR_Ai; or an equimolar mixture of pVaxPR_Ai2mut and pVaxPR_AI3mut (DR PRi mix); pVAX1; or PBS (series III, Table 1). Injections were delivered in the hips and one shoulder, rendering a 50% higher dose of total DNA compared to series I and II (Table 1). The general condition of the animals corresponded to the physiological norm throughout the whole study period (Appendix A). There were no significant differences in body mass between the recipients of any of the plasmids and the control group receiving PBS (Appendix A). We also performed an analysis of complete blood counts and blood biochemistry in all animals on 1 and 13 days post-plasmid-boost (Table 2 and Table 3).

The analysis revealed a difference between the groups in white-blood-cell (WBC) counts (WBC: F(3.19) = 5.2262, *p* = 0.0084; KW-H(3.23) = 10.352, *p* = 0.0158). The mice receiving the DR PRi mix had lower WBC counts than those receiving PR_Ai or pVAX1 (*p* = 0.009, MW test), but not the PBS mice (Table 2). Furthermore, mice receiving the DR PRi mix had a larger population of lymphocytes (LYM) than the control mice receiving PBS (LYM:F(3.19) = 3.22, *p* = 0.0459; KW-H(3.23) = 9.8235, *p* = 0.0201) (Table 2). In mice immunized with the DR PRi mix, LYM counts were higher than in mice receiving PR_Ai (*p* = 0.002) or PBS (*p* = 0.009) and had a weak tendency to be higher than LYM counts in the pVAX1 group (*p* = 0.12). Additionally, in mice receiving the DR PRi mix, granulocyte (GRAN) counts were significantly lower than in the PBS group and tended to be lower than in the PR_Ai and vector groups (Table 2). Thus, immunization with the DR PRi mix preserved the total WBC counts within the range observed in the control-PBS-immunized mice, but induced an increase in the LYM subpopulation reflecting their activation. The DR PRi mix group also demonstrated decreased (statistically significant or as a tendency) granulocyte counts compared to the other groups.

Pairwise comparisons did not reveal statistically significant long-term differences between the study groups in other parameters of the blood formula, or bone marrow composition (Appendix A) (Table 2; Appendix A). Variations in red blood cell and platelet counts and metamyelocyte population in the bone marrow were limited to day 1 post-booster-immunization (Table 2). We also registered an increase in the mass of the thymus related to body mass in mice receiving PR_Ai, while no increase was observed in mice receiving the PR DRi mix (Table 2). The latter, on the contrary, had a significantly decreased relative mass of the spleen (both by day 12 if compared to the group receiving PBS; Table 2).

The analysis of biochemical parameters revealed a difference between the groups in the levels of triglycerides (TGC) (F(3.16) = 3.2437, *p* < 0.05). TGC level serves as a biochemical marker of the size of the leukocyte population and is positively associated with leukocyte counts [61]. Here, as well, TGC levels significantly correlated with the size of the lymphocyte population in the bone marrow (R = 0.45, *p* < 0.05 Spearman). Mice receiving the DR PRi mix had or tended to have higher TGC levels in the blood than the other mouse groups: their TGC levels tended to be higher than in the PR_Ai mice (*p* = 0.09 MW), and were higher than in the PBS mice (*p* = 0.03), supporting immune activation. 

In addition, mice receiving the DR PRi mix had lower levels of urea in the blood than the other mouse groups (urea: F(3.16) = 3.4753, *p* = 0.0409; KW-H(3.20) = 8.2854, *p* = 0.0405). We also observed a weak correlation of the levels of urea and albumin (ALB) (urea: ALB: r = 0.2833, *p* = 0.07650; r2 = 0.0803) (Appendix A). Levels of urea and ALB characterize the filtration function of the kidneys [62]. Disturbance of the latter (e.g., nephrotic syndrome and uremia) affects the levels of serum lipids/lipoproteins due to alterations in lipid metabolism [62,63]. In our case, the levels of urea and ALB were found to correlate with the levels of TGC (urea:TGC: r = 0.4279, *p* = 0.00590; ALB:TGC: r = 0.3148, *p* = 0.04790). However, in all mice, fluctuations in both urea and TGC levels were within the normal range recorded for BALB/c mice (7.14 (2.5–11.1) mmol/L for urea, and 3.4 (1.14–6.7) mmol/L for TGC (mean; 95% CI). ALB levels were increased over the norm (37 (31–53) g/L, mean; 95% CI) in two mice only (one receiving the PR DR mix and one in the pVAX1 group) with no significant difference between the groups at any of the time points (for normal values, see https://www.criver.com/sites/default/files/resources/BALBcMouseClinicalPathologyData.pdf accessed on 20 January 2022). The mice receiving PR_Ai had lower hematocrit (HCT), hemoglobin (HGB), and cholesterol levels compared to pVAX1, but not PBS mice; the difference disappeared by day 12 after the boost (Table 3). Other parameters of the biochemical blood analysis did not differ (Table 3; Appendix A). Altogether, biochemical blood parameters revealed no toxic effects of the PR_A gene variants, also with respect to the filtration function of the kidneys.

Deviations from the norm in all above parameters were observed in single mice with no difference in occurrence between the groups. Thus, the “wild-type” as well as drug-resistant PR variants delivered by DNA immunization with EP via the optimal immunization scheme exerted no immunotoxic effects in the recipient mice.

The results of immunotoxicity tests demonstrated the favorable safety profile of PR_A variants. Interestingly, an increased relative mass of the thymus is associated with lymphoid thymic hyperplasia due to inflammation [64], while a decrease in the relative mass of the spleen accompanies the migration of splenocytes into the blood/chemotaxis indicative of lymphocyte proliferation [65,66]. A shift to one or the other scenario agrees with the fluctuations in WBC and their GRAN and LYM subpopulations and TGC levels in the blood, with DNA immunization with the empty vector followed by electroporation causing inflammation, and DNA immunization with DR PRs shifting it towards the development of cellular immune response.

#### 5.4.4. Assessment of Protective Capacity of PR Immunization: Growth of Tumor Cells Expressing PR Can Be Prevented Only by Immunization with PR Gene Identical to That Expressed by Tumor Cells

To test the protective potential of PR_A variants, BALB/c mice were immunized with plasmids encoding PR_Ai, PR_Ai2mut, and PR_Ai3mut via a prime–boost regimen (Table 1, series IV) and then challenged with 4T1luc2 cells expressing PR_A3i (PR20.2 cells). The control mice were either mock-immunized with PBS and challenged with PR20.2, or immunized with pVAX1 and challenged with the parental 4T1luc2 cells (Table 1, series IV). Tumor growth was monitored by calipering. Due to the expression of luciferase by parental cells and the PR20.2 clone, tumor growth could be measured using bioluminescent imaging (BLI) (examples of BLI images are given in Appendix A). DNA immunization with PR_Ai or the empty vector did not lead to any notable changes in the growth of either PR20.2 or parental 4T1luc cells (Figure 7A–D). The monitoring of BLI from injection sites demonstrated that DNA immunization with PR_Ai3mut coding for the same PR variant as expressed by PR20.2 cells significantly delayed tumor growth (Figure 7A,C). Furthermore, by the experimental endpoint, a palpable tumor was formed only in one mouse from this group, in one injection site (totally, 1/10 sites). The size of this tumor was 15 times lower than the average tumor size in other groups (13 mm^2^ compared to 279 mm^2^, Figure 7D). The BLI signal from the sites of injection of tumor cells in mice DNA-immunized with PR_Ai2mut did not differ from that in PBS-immunized mice receiving PR20.2 (Figure 7A), but the tumors in PR_Ai2mut animals were larger than in the PBS controls (*p* < 0.05, Figure 7D).

By the experimental endpoint, all mice were euthanized, and the areas of injections were excised, fixed, paraffin-embedded, sectioned, stained with H&E, and assessed by an independent pathologist. PR20.2 cells induced the formation of primary epithelial tumors with a spindle-shaped cell solid architecture (Figure 7E, panel 1). Tumors were typically highly aggressive, high-grade (Grade 3) adenocarcinomas consisting of polymorphic cells with high mitotic activity (Figure 7E, panel 2, red arrows) and a weakly preserved glandular structure with stromal desmoplasia (Figure 7E, panels 3 and 4, respectively). The study of tumor sections revealed the presence of large necrotic areas (Figure 7E, panels 5, 6 encircled). The tumors formed by the parental 4T1luc2 cells in the control mice were identical to those described by us and by others previously [50,52].

The infiltration of tumor cells into mouse organs characterizing the motility of tumor cells in vivo was assessed using BLI of mouse organs at the experimental endpoint as was described by us earlier [49]. The most affected organs were the lungs and to a lesser extent the liver; spleen was not involved (Figure 8A–C, Appendix A). In infiltration into the lungs, PR20.2 tended to be more aggressive than the parental 4T1luc2 cells (*p* = 0.06). Neither immunization with PR_Ai or PR_Ai3mut, nor with the empty vector, had any effect on the migration of tumor cells into the lungs (no difference from PBS treatment, *p* > 0.1) (Figure 8A). Interestingly, photon flux from the lungs in the PR_Ai- or PR_Ai3mut-immunized groups was on the background level (8.3 × 10^3^ ± 3.8 and 8.3 × 10^3^ ± 2.2, respectively) indicating the absence of tumor cells in the organ (one 4T1luc2 cell releases 6500 photons per sec; http://www.caliperls.com/assets/014/7158.pdf, assessed on 3 August 2022). On the contrary, photon flux from the lungs of the PR_Ai2mut-immunized mice was increased, indicating an enhanced migration of PR20.2 cells into the lungs compared to both treatment with PBS (*p* = 0.018) and PR_Ai and PR_Ai3mut DNA immunization (*p* = 0.008 for both pairs) (Figure 8A). The average BLI signal of 21,000 ± 6000 p/s in the latter groups significantly exceeded the background level, indicating the presence of at least three tumor cells in the lungs of each mouse.

BLI from the liver and spleen indicated an absence or very low numbers of infiltrating tumor cells in both mice implanted with PR20.2 and with the parental cell line (Figure 8B,C), significantly lower than in the lungs (Friedman ANOVA and Kendall coeff. of concordance: ANOVA chi sqr. (N = 9, df = 1) = 5.444444 *p* = 0.01963; coeff. of concordance = 0.60494 aver. rank r = 0.55556) (Appendix A). Of note, in 60% (3/5) of the mice immunized with PR_Ai2mut and challenged with PR20.2, the BLI signal from the liver exceeded the background level (the average photon flux from the liver in these groups exceeded 8000 p/s; Figure 8B), indicating the presence of tumor cells in the liver.

Thus, immunization with PR_Ai3mut prevented the infiltration of 4T1luc2_PR20.2 cells into the distal organs, specifically the lungs. Little or no infiltration was observed in PR_Ai-immunized mice. On the contrary, immunization with PR_Ai2mut strongly enhanced the motility of PR20.2 cells, specifically the infiltration into the lungs typical of 4T1 cells [52] and also into the liver.

Livers were fixed, paraffin-embedded, sectioned, stained with H&E, and assessed by an experienced pathologist with the registration of the following parameters (in numbers): metastases formed by tumor cells, metastases formed by tumor cells and neutrophils (“mixed metastasis”), neutrophil infiltrates in the absence of visible tumor cells, and the size of these formations. This assessment confirmed that mice implanted with PR20.2 cells had metastases in the liver. These were mainly mixed metastases consisting of tumor cells and neutrophils (Figure 9(A1)), often micrometastasis (Figure 9(A2)). The metastases were widely spread (Figure 9(A3)). The metastatic tumor cells invaded the blood vessels (Figure 9(A4)) and infiltrated into the bile duct (Figure 9(A5)). Some metastases consisted of just tumor cells (Figure 9(A6)). The structures were similar to that shown earlier in the livers of mice implanted with 4T1 and 4T1luc2 cells [50]. No metastases were found in the livers of mice DNA-immunized with PR_Ai3mut (Figure 9B). Immunization with pVaxPR_Ai or pVAX1 had no effect on either the number or the size of metastases formed by tumor cells used in challenge (PR20.2 in case of PR_Ai, and 4T1luc2 in case of pVAX1 immunization) (Figure 9B,C). On the contrary, DNA immunization with PR_Ai2mut led to an increase in the number and the size of mixed metastases compared to these parameters in the control group (Figure 9B,C) while no difference was found in the infiltration of neutrophils or tumor cells alone (Appendix A). Thus, only immunization with the PR_A variant identical to that expressed by tumor cells, namely PR_Ai3mut, protected mice from the invasion of distal organs by tumor cells with the formation of metastases, as evidenced by ex vivo BLI and histochemical assessment.

The splenocytes of DNA-immunized and control mice challenged with PR-expressing and parental 4T1luc2 cells collected at the experimental endpoint were assessed for the ability to secrete IFN-γ/IL-2 in response to stimulation with PR-derived peptides using Fluorospot. No PR-specific response was detected in PBS-immunized mice challenged with PR20.2, indicating that the implantation of PR-expressing tumors as such did not generate any PR-specific immunity. The immune response to PR peptides in PR_Ai and PR_Ai2mut mice was weak-to-undetectable, whereas mice DNA-immunized with PR_Ai3mut developed a strong immune response to peptides A1–15 and A76–90, represented by a high number of IFN-γ-, IL-2-, and IFN-γ/IL-2-producing cells (Figure 10A–C). Importantly, the number of splenocytes secreting IFN-γ, IL-2, and IFN-γ/IL-2 in response to stimulation with peptides A1–15 and A76–90 was inversely correlated with the total tumor volume (Table 4). Of note, mice DNA-immunized with PR_Ai3mut, but not other animals, had higher overall splenocyte activation in response to stimulation with mitogen concanavalin A (ConA), indicating the preservation of T-cell viability/functions (Appendix A).

## 6. Discussion

The aim of this study was to design DNA immunogens against drug resistance (DR) in HIV infection based on the protease (PR) of the HIV-1 FSU-A strain, and characterize their expression, the activity of the encoded proteins, the immunogenic performance in mice, and protective potential in a murine model of HIV-1-challenge-exploiting tumor cells expressing the immunogen. The secondary aims included the identification of the best therapeutic vaccine candidate, as well as gaining knowledge in the mechanisms determining the immunogenicity of PR-encoding plasmids. Earlier, promising results demonstrated the feasibility of DNA vaccines based on synthetic genes encoding consensus viral proteins [67,68,69,70]. The design of such a consensus here was simplified by the low variability of the FSU_A strain [6,11,71]. Indeed, consensus sequences generated from two sets of sequences, one dated 1997–2003 (n = 206) and the other dated 2008–2015 (n = 120), were identical, confirming low genetic diversity and potential applicability of the consensus gene as an HIV DNA vaccine platform.

The designed PR_A genes encoded the desired protein variants; however, the intracellular levels of all variants were very low. The low expression of enzymatically active PR variants is attributed to their enzymatic activity. PR is a pepsin-like aspartic protease [72,73]. It cleaves HIV-1 gag and pol resulting in their maturation, but also performs an autocleavage at L5-W6, L33-E34, and L63-I64 [74]. Autocleavage is responsible for the fast degradation of the protein in the cells and in recombinant protein preparations [74]. In addition, protease activity is toxic to the expressing cells [75]. In eukaryotic cells, PR activity causes a concentration-dependent decrease in the mitochondrial membrane potential, the activation of caspase 9, PARP cleavage, and DNA fragmentation [76]. Mutations that inhibit autocleavage without affecting the enzymatic activity were identified, but they never occur in nature (positions 7, 33, and 63; [77]). Naturally selected are amino acid residues promoting the autocleavage. We also could see a 100% conservation of Q7, L33, and L63 in our alignment of PR_A sequences, i.e., the protease of all HIV-1 clades, including clade A, has evolved as a short-living protein, destined to commit suicide after the completion of HIV-1 gag/pol cleavage, in order to reduce the direct toxicity to the infected/expressing cells.

Logically, the inactivation of PR_A should have led to the inhibition of autocleavage with the increase in intracellular content of the protein. The protease-inactivating mutation D25N was previously shown to significantly increase the level of expression of PR of HIV-1 clade B [41]. We were able to demonstrate this only for PR_Ai3mut; the intracellular levels of the other two PR variants were still low despite the introduction of D25N. The primary DR mutations decreased the enzymatic activity of the protein (shown for DR mutations in aa positions 36, 54, and 82 of PR; [78]) and could be expected to increase the intracellular levels of mutated proteases, similar to the effect of D25N. Instead, the introduction of M46I/I54V mutations decreased the intracellular levels of the mutants compared to the parental PR_A. At the same time, the intracellular content of five PR_A variants (all except stable PR_Ai3mut) was increased after the treatment of expressing cells with the proteasomal inhibitor MG132. Proteasomal inhibitors interfere with the processing, release, and maturation of HIV-1 gag polyprotein, but they do not affect the enzymatic activity of PR [60]. Thus, fluctuations in the intracellular levels of PR variants after inactivation and MG132 treatment could not be attributed to the inhibition of PR autocleavage.

An alternative explanation takes into account protein sensitivity to cellular proteases. Indeed, while treatment with MG132 increased the intracellular content of PR_A2mut by 20 times, it had no effect on the expression of the initially highly expressed PR_Ai3mut. Earlier studies observed that the introduction of certain DR mutations change the conformation and thermal stability of PR [79,80], resulting in low stability and the sensitivity to cleavage by cellular proteases [81]. The observed effect of MG132 could be explained in terms of proteolytic stability, by the inhibition of PR degradation by the chymotrypsin-like activities of the proteasome. In this scenario, mutations D25N and V82A collaborate to prevent proteasomal degradation.

Next, we analyzed the immunogenicity of the consensus HIV-1 FSU_A PR_A and its inactive variant PR_Ai. Both PR_A and PR_Ai were strongly immunogenic. The immune response induced by PR variants was cross-reactive with respect to clade-specific peptides. Mice developed no immune response against aa 31–56 or 56–70. Their immunodominant epitopes were localized at aa 1–15 and aa 71–90 which contain known T-cell epitopes recognized in humans [82]. Importantly, mice immunized with the “wild-type” PR 82V variants recognized peptides with 82V, but not those with DR mutation 82A, indicating that aa residue 82 was essential for the recognition of protease by specific T cells. 

The high immunogenicity of the consensus-inactivated PR_Ai allowed us to use it as a platform for the design of immunogens targeting drug-resistant (DR) PR variants. The mutations we selected were primary DR mutations evolving in the HIV-1 protease of clade A in response to drugs applied in the territory of FSU_A circulation, namely M46I, I54V, and V82A [54] (Stanford HIV Drug Resistance Database. at http://hivdb.stanford.edu/ accessed on 18 May 2018). Our in silico study predicted these DR mutations would not cause any significant change in protein processing or the affinity of epitope binding to MHC class I or MHC class II. 

The in vitro observations of the difference in proteasomal stability/degradation indicated that PR_Ai2mut and PR_Ai3mut would have distinct immunogenicity profiles. In the case of PR_Ai2mut, enhanced proteasomal degradation would generate more peptides for presentation in the context of MHC class I providing for the higher cellular immunogenicity of the PR M46I/I54V variant. This, however, would not be the case for PR_Ai3mut insensitive to the inhibitor of proteasomal degradation. This scenario was tested in BALB/c mice. As we expected, PR_Ai2mut was highly immunogenic. It induced a stronger cellular response against the immunodominant epitopes at aa 1–15 and aa 71–90 of PR than the other two other variants, and was highly immunogenic already after the DNA prime. Thus, the M46I and I54V mutations enhanced protease immunogenicity, although the change did not involve the immune recognition of the mutations. Additionally, as we expected, PR_Ai3mut with M46I/I54V/V82A had comparatively low immunogenicity, with the immune response limited to the epitopic region at aa 71–90 of the PR. 

To characterize the specificity of the immune response against DR mutations, we used peptides representing PR regions aa 31–56, 56–70, 71–85, and 76–90 with and without DR mutations. Mutations M46I and I54V had no effect on the specificity of the anti-PR immune response in mice; the respective peptides (either with or without mutations) were not recognized, but, interestingly, they enhanced the immunogenicity of the “carrier” PR variant. On the contrary, the mutation at aa residue 82 had a marked effect on the immune response against aa 71–90. The region between aa 75 and 84 was earlier shown to contain an immunodominant epitope of protease for HLA-A0201 [83] coinciding with the dominant epitope recognized in H2Dd and HLA-A0201 transgenic mice [41]. Interestingly, all DR and non-DR-PR variants induced an immune response against nonmutated peptide aa 75–84, indicating that the aa residue in position 82 does not interfere with the immune recognition. On the contrary, mice receiving PR_A 82V variants poorly recognized A76–90dr, while mice receiving PR with 82A recognized both A76–90 and A76–90dr. This could be explained if one suggests that the region aa 75–90 of PR contains two epitopes: one at the N-terminus at aa 75–84 recognized by the immune system of all PR gene recipients irrespective of the nature of the aa residue in position 82; and the other, shifted to the C-terminus, with V82A crucial for both immunogenicity and immune recognition.

From this analysis of epitopic specificity, we moved further to the type of the responding T cells determined by flow cytometry with intracellular cytokine staining (ICCS). Cellular immune response against PR was dominated by CD8+ T cells. Flow cytometry with ICCS confirmed the immunogenic characteristics of PR variants obtained by Fluorospot. PR_Ai and PR_Ai2mut exhibited similar immunogenic profiles, despite the presence of two DR mutations in the latter. PR_Ai3mut was different. Interestingly, on the contrary to the Fluorospot data on the equal immune recognition by PR_Ai3mut recipients of the peptides with 82V and 82A, the CD8+ T cells of mice receiving PR_Ai3mut could not recognize the nonmutated peptides AB71–85 and A76–90. Instead, they solely recognized the peptides with DR mutations AB71–85dr and A76–90dr, while no recognition of the latter peptides was registered for the CD8+ T cells of mice receiving PR_Ai or PR_Ai2mut. Thus, on the level of CD8+ T-cell response, PR_Ai and PR_Ai2mut had similar immunogenicity, and PR_Ai3mut with the 82A mutation was different, confirming in vitro data on the sensitivity of PR variants to cleavage by the proteasome. Importantly, only PR_Ai3mut was able to induce a CD8+ T-cell immune response recognizing V82A.

According to flow cytometry, the CD4+ T-cell response was low-to-undetectable (Appendix A). The low CD4+ T-cell response falls in line with the earlier findings [41] including the absence of a PR-specific antibody response. However, flow cytometry was carried out on frozen cells after thawing. Freezing/thawing may lead to a significant loss of viable CD4+ cells, while CD8+ T cells are not affected [84], which would explain why T-cell reactivity registered by flow cytometry exclusively involved CD8+ T cells. As noted above, in the capacity to induce a CD8+ T-cell response, PR_Ai and PR_Ai2mut did not differ; however, the assessment of the integral T-cell response carried out by Fluorospot on fresh cells demonstrated that PR_Ai2mut was more immunogenic. This difference may relate to the increased immunogenicity of PR_Ai2mut observed in Fluorospot to the reactivity of specific CD4+ T cells (lost in tests carried out on thawed cells).

Finally, we assessed the functionality of the induced immune response. For this, we challenged mice DNA-immunized with PR variants with tumor cells expressing HIV-1 protease. Through lentiviral transduction, we obtained a subclone of murine mammary gland adenocarcinoma 4T1luc2 cells stably expressing the PR_Ai3mut variant (dubbed PR20.2) and luciferase, which allowed us to perform the bioluminescent imaging (BLI) of tumor growth in vivo. Other active and inactive PR variants were toxic to expressing cells, with the respective lentiviral variants not forming viable particles, not allowing us to perform challenge tests with the respective PR variants. DNA immunization with the PR_Ai3mut-encoding PR variant identical to that expressed by PR20.2 cells resulted in nearly complete protection against tumor challenge. Mice DNA-immunized with PR_Ai3mut had no infiltrating tumor cells in distal organs detectable by ex vivo BLI, and no metastasis in the liver according to histochemical assessment. Importantly, they retained functional cellular responses demonstrated by the capacity of their splenocytes to respond to activation by mitogen ConA, and exhibited a specific cellular response against PR aa 1–15 and also aa 76–90 correlated with protection. 

On the contrary, DNA immunization with PR_Ai2mut was shown to be more immunogenic, and led to the formation of larger tumors than control PBS-immunized mice. Additionally, they demonstrated increased infiltration of tumor cells into the distal organs (livers and lungs). Histopathological evaluation of their organ sections revealed metastases of tumor cells surrounded by neutrophil infiltrates. No changes in the parameters of tumor growth or metastatic activity were observed for the parental tumor cells in mice DNA-immunized with an empty vector compared to the control mice receiving PBS. Thus, enhancement of the growth rate and of the in vivo motility of PR20.2 cells in PR_Ai2mut immunized mice was not related to an immune activation induced by plasmid DNA as such, but reflected a specific effect of PR_Ai2mut as the immunogen.

We attempted to dissect the nature of this effect. T cells of mice DNA-immunized with PR_Ai2mut recognized peptides spanning aa 71–90 both with and without DR mutations. This reactivity was explicitly demonstrated by the IFN-γ/IL-2 Fluorospot, while the assay of their CD8+ T cells showed no specific mono IFN-γ or IL-2 production in response to either the peptide covering aa 71–84 or aa 76–90 attributing the T-cell reactivity to peptides spanning the region of aa 71–90 to CD4+ T cells. In this scenario, the PR20.2-expressing tumors formed in PR_Ai2mut-immunized mice could be infiltrated by CD4+ T cells inefficient in or incapable of killing tumor cells expressing the PR V82A mutant, but causing local inflammation and enhancing tumor growth. Selective accumulation in the tumors of CD4+ T cells (subpopulation of Tregs) was described for murine fibrosarcomas [85]. At the late stage of tumor progression, these cells constituted the majority of tumor-infiltrating lymphocytes, concealing the immunogenicity of tumor cells and allowing for the progressive growth of antigenic tumors [85]. It has also been proposed that the tumor microenvironment can convert the effector CD4+ T cells beneficial for anti-tumor immunity into Tregs causing the additional suppression of anti-tumor immunity [86]. Likewise, in PR_Ai2mut-immunized mice, the immune recognition of aa 74–85 by CD4+ T cells without effector/with Treg activity (or turned into Tregs after infiltration) could have enhanced tumor growth and increased the metastatic activity of PR20.2 cells. In contrast to this “CD4+ T scenario”, in mice immunized with PR_Ai3mut presenting the V82A mutation, the immune response was dominated by CD8+ T cells specific to T-cell epitope(s) at aa 76–90 recognizing the V82A mutation, which were able to prevent the growth and metastatic activity of tumor cells despite the comparatively low specific T-cell response.

Overall, our data demonstrate that the candidate DNA vaccine against drug resistance in HIV-1 infection does not benefit from the inclusion of all known DR mutations, but only those that are associated with changes in antigenicity and immunogenicity. The development of such a vaccine requires localization in the regions with DR mutations of the epitopes of immune cells with lytic activity, preferably CD8+ T cells, omitting DR mutations that lie outside of the epitopes and/or inducing an immune response with no lytic activity. We also demonstrated the functionality of a novel approach to test the protective potential of such immunogens by challenging immunized mice with tumor cells expressing the respective DR HIV proteins. The need in such a system is well-illustrated by the fact that the immunogenicity of DR PR variants could not be predicted by in silico assessments of MHC class I immunogenicity, and their protective effect did not directly follow their immunogenicity. 

Our data support previous observations that DR mutations may lead not only to an escape from antiretroviral drugs, but also to an immune escape. This phenomenon was demonstrated for DR HIV-1 enzymes: protease in relation to mutations V82A [83], I47A, and G48V [82] and reverse transcriptase in relation to DR mutations E138G/A/K [87]. A virus with these mutations would have a double advantage compared to the wild-type variant, and an escape from the drug and from the host immune response, as confirmed by the study demonstrating that the presence of certain alleles affects the frequency of DR mutations [88]. Mutations allowing for the dual drug and immune escape would have less probability of reversal after cessation of the therapy and will pertain after viral transmission to the new, yet untreated, host. This would stimulate the increased circulation/spread of drug-resistant viral variants. 

It would be vitally important to have approaches to hinder spread of such DR HIV-1 variants, alongside the prevention or hindering of viral evolution towards drug resistance. Drug resistance and immune escape are not a one-way/one-direction route. While M46I and I47A in PR impair or abolish CTL recognition in some of the patients, variants with M46L and I47V are well-recognized in nearly all, with the load of DR HIV-1 significantly lower in patients that had a CTL response against respective variants [82]. Another DR mutation in protease, L90M, was shown to occur at lower frequencies in hosts that harbor the B*15, B*48, or A*32 human leukocyte antigen subtypes that recognize a novel epitope formed by L90M [25]. Both examples demonstrate the potential and the utility of the immunogens which would induce an immune response against DR mutations in HIV-1 patients on ART. Our study shows that DNA immunization is capable of inducing such a response and that this response is protective in a mouse model.

## 7. Conclusions

We have shown that an immune response induced by DNA immunization with a drug-resistant variant of HIV-1 protease is able to protect mice against the growth and metastatic activity of tumor cells expressing the respective drug-resistant PR variant. This is a proof of concept that DNA immunization can induce a cytolytic immune response recognizing single amino acids including mutations of drug resistance and immune escape, promoting its application in combination with ART to hinder the development of drug resistance in HIV-1 infection. The finding that a single amino acid mutation in the immunogen can define its ability to prevent the growth of antigen-expressing tumor cells is also crucially important for the development of cancer vaccines.

## Figures and Tables

**Figure 1 cancers-15-00238-f001:**
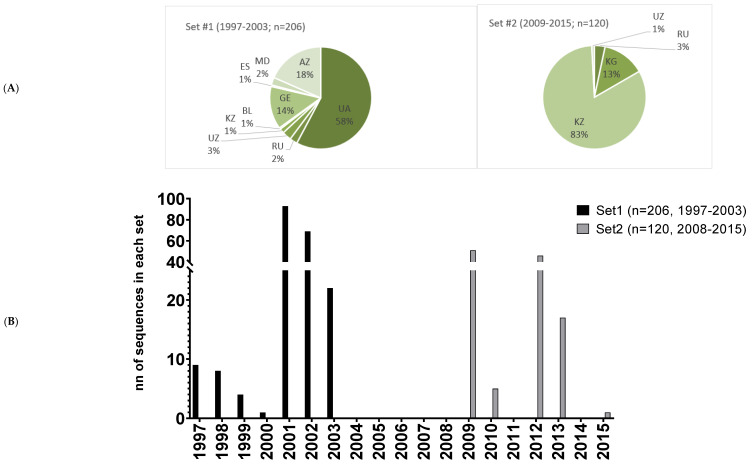
Design of amino acid consensus of protease of the HIV-1 FSU-A strain (PR_A) and its variants with drug resistance (DR) mutations. Geographical representation (**A**) and time distribution (**B**) of entries constituting two datasets of amino acid (aa) sequences of PR_A used to generate the PR_A consensus; alignment of the consensus amino acid sequence of PR_A, and PR_A variants with mutation of inactivation D25N (PR_Ai), D25N and DR mutations M46I/I54V (PR_Ai2mut), and D25N and DR mutations M46I/I54V/V82A (PR_Ai3mut), and synthetic peptides chosen to study the cellular immune response against protease (**C**). Amino acid sequence of PR of HIV-1 clade B isolate HXB2 (PR_B) is given above the PR_A consensus for comparison. Sequences in Sets #1 and #2 were aligned using MUSCLE software (www.ebi.ac.uk/Tools/msa/muscle/ accessed on 15 May 2018); processing of the data using Geneious 8.1.2 generated one and the same sequence. Mutation D25N abrogating protease activity was described earlier [30]. Primary DR mutations were selected based on the data published in the HIV Stanford resistance database. Synthetic peptides represent regions homologous to the known B- and T-cell epitopes in HIV-1 PR listed in the Los Alamos HIV-1 and IEDB databases. The prefix before the peptide depicts the specificity to subtype A (**A**), subtype B (**B**), or conserved regions identical in PR_A and PR_B (**A**,**B**). Alignment shows only aa residues different from that in the consensus PR_A.

**Figure 2 cancers-15-00238-f002:**
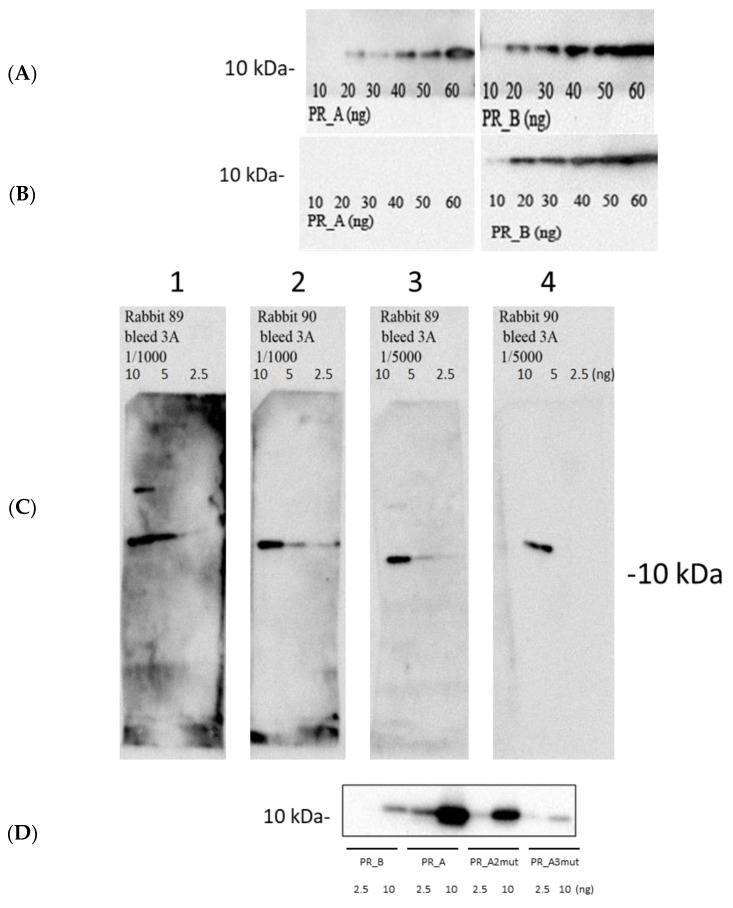
Characterization of the variants of consensus protease of HIV-1 FSU_A strain (PR_A) with and without drug resistance (DR) and inactivation mutations expressed in *E. coli*. Western blotting of recombinant PR_A taken in amounts of 10 to 60 ng and recombinant PR of HIV-1 clade B HXB2 strain (the amount of protein loaded into the well is depicted below the lanes) with polyclonal rabbit anti-protease antibodies (Fitzgerald, GA, USA) (**A**) or monoclonal anti-protease clade B antibodies clone 1696 (Exbio, Vestec, Czech Republic) (**B**); Western blotting of PR_A taken in amounts of 2.5, 5, and 10 ng per well with sera of PR_A-immunized rabbits #89 ((**C**), panels 1 and 3) or #90 ((**C**), panels 2 and 4) diluted 1:1000 (C1, C2) or 1:5000 (C3, C4); Western blotting of recombinant PR_B, PR_A, and PR_A variants with mutations M46I/I54V (PR_A2mut) and M46I/I54V/V82A (PR_A3mut) in amounts of 2.5 and 10 ng (as depicted below the lanes) with anti-PR_A rabbit sera #89 diluted 1:1000 (**D**); Activity of the recombinant proteases PR_A, PR_A2mut, and PR_A3mut taken in an amount of 10 ng measured using a FRET assay (SensoLyte 490, Anaspec, CA, USA), represented as % of the activity of PR_B; data represent the average of the triplicate test with SD (**E**).

**Figure 3 cancers-15-00238-f003:**
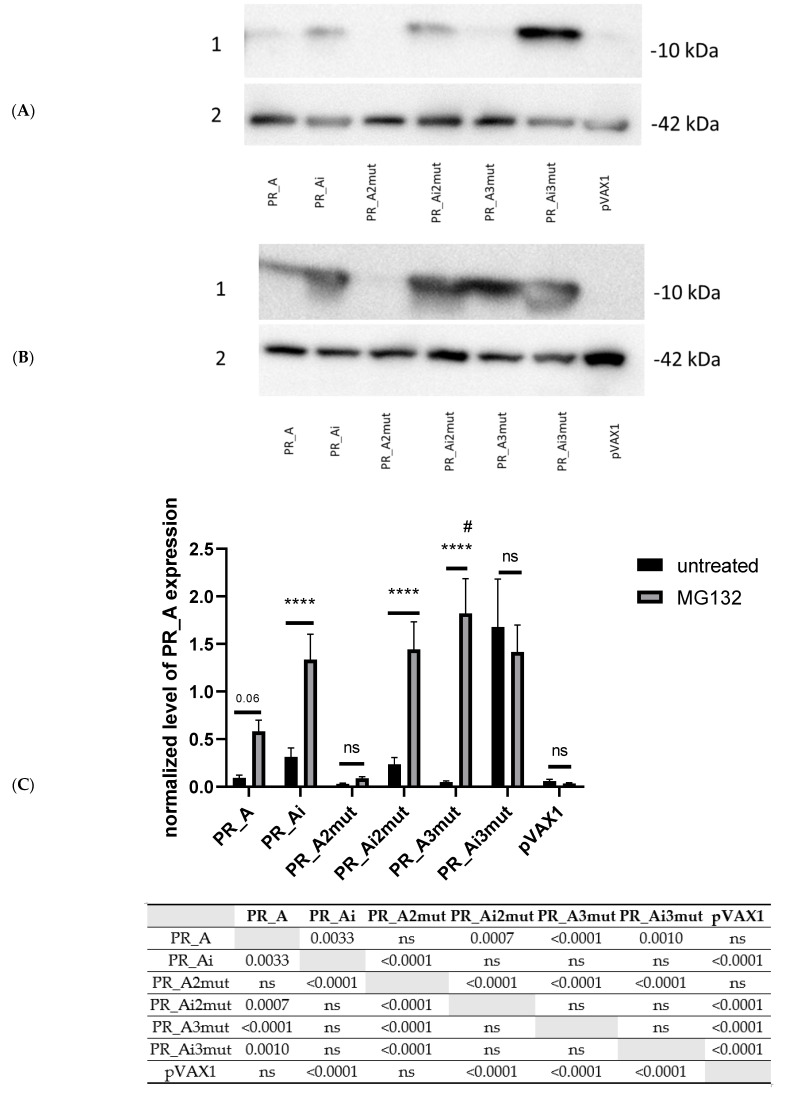
The effect of drug resistance (DR) and inactivation mutations on the intracellular levels of HIV-1 FSU_A consensus protease (PR_A) variants in eukaryotic cells. Western blotting of lysates of HeLa cells transfected with plasmids pVaxPR_A, pVaxPR_Ai, pVaxPR_A2mut, pVaxPR_Ai2mut, pVaxPR_A3mut, pVaxPR_Ai3mut, and pVAX1 not treated (**A**), or treated for the last 18 h with MG132 (5 µM) (**B**); Relative expression of PR_A variants normalized to the expression of actin (**C**). The conditions of cell culture are described in the Materials and Methods. Western blots were first stained with polyclonal anti-PR-A antibodies (rabbit #89; 1:4000) (A-1, B-1) then stripped and restained with monoclonal anti-actin antibodies (A-2, B-2). The content of protease variants was assessed with ImageJ. Results of at least two independent runs. * depicts a significant difference in the intracellular content of PR variants depending on MG132 treatment, ns: Insignificance is denoted by ns, **** *p* < 0.0001. # depicts a significant difference in the intracellular content of PR variants depending on mutations without treatment (*p* < 0.0001). The table underneath indicates significant differences in the intracellular content of PR variants depending on mutations after MG132 treatment and two-way ANOVA with Sidak’s and Tukey’s multiple comparisons correction.

**Figure 4 cancers-15-00238-f004:**
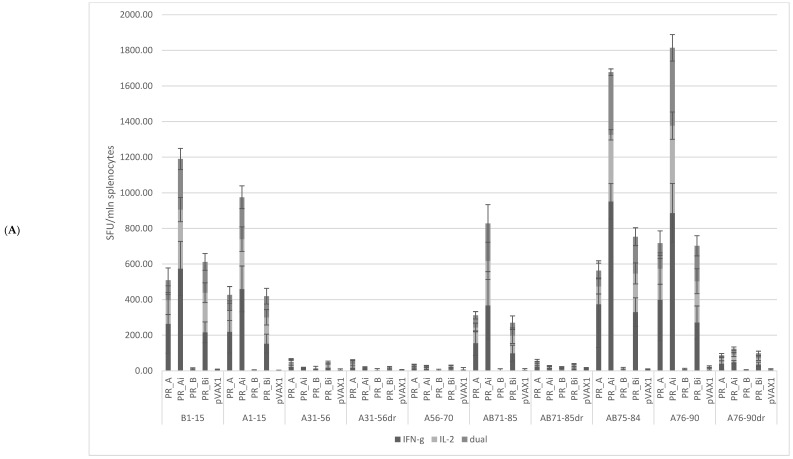
IFN-γ, IL-2, and dual IFN-γ/IL-2 secretion of splenocytes of BALB/c mice DNA-immunized with consensus protease of HIV-1 clade A FSU_A strain (PR_A), PR of HIV-1 subtype B (PR_B), their variants inactivated by D25N mutation (PR_Ai and PR_Bi, respectively), or empty vector pVAX1 (series I, Table 1) after in vitro stimulation with HIV-1 PR-derived peptides (Figure 1C). Pile-up of the number of IFN-γ-, IL-2-, and dual IFN-γ/IL-2-secreting cells registered as spots per mln splenocytes (SFC/mln) in response to in vitro stimulation with PR_A-derived peptides (**A**); analysis of the difference in secretion of IFN-γ (panel 1), IL-2 (panel 2), or dual secretion of IFN-γ/IL-2 (panel 3) in response to splenocyte stimulation with peptides common for PR_A and PR_B variants (**B**), clade-specific peptides (**C**), and peptides encompassing regions of drug resistance (DR) mutations with peptides bearing DR mutations dubbed as “dr” (**D**). Cytokine production was assessed by a dual IFN-γ/IL-2 Fluorospot. Data are represented as the net SFC-expressing IFN-γ, IL-2, and dual IFN-γ/IL-2 per 10^6^ splenocytes, with error bars, SD. Statistics were carried out using a two-way ANOVA with Tukey’s multiple comparisons correction, * *p* < 0.05, ** *p* < 0.01, *** *p* < 0.001; **** *p* < 0.0001.

**Figure 5 cancers-15-00238-f005:**
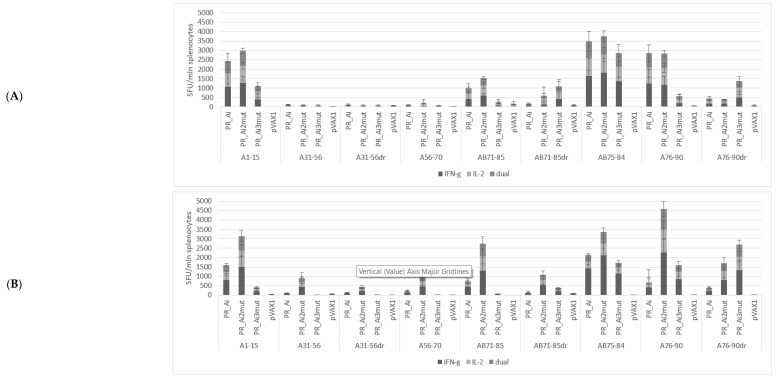
Frequencies of IFN-γ-, IL-2-, and dual IFN-γ/IL-2-secreting splenocytes in BALB/c mice primed and boosted with plasmids encoding inactivated consensus protease of HIV-1 clade A FSU_A strain (PR_Ai), its variants with M46I/I54V (PR_Ai2mut) and M46I/I54V/V82A (PR_Ai3mut) mutations, or empty vector pVAX1 upon stimulation with HIV-1 PR-derived peptides (series II, Table 1). IFN-γ, IL-2, and dual IFN-γ/IL-2 secretion by splenocytes PR_Ai, PR_Ai2mut-, PR_Ai3mut-, or empty vector pVAX1-immunized mice in response to in vitro stimulation with synthetic peptides (Figure 1C) performed 21 days after the prime ((**A**,**C**) panels 1–3) and 21 days after the boost ((**B**,**D**) panels 1–3); Comparison of anti-PR immune response by splenocytes of mice immunized with PR_Ai (I), PR_Ai2mut (II), and PR_Ai3mut (III) after the prime and after the boost (**E**). Panels depict the secretion of IFN-γ (C1, D1, and E1), IL-2 (C2, D2, and E2) and IFN-γ/IL-2 (C3, D3, and E3). Cytokine secretion was assessed by a dual IFN-γ/IL-2 FluoroSpot. The data represent net spot-forming cells (SFC) per million cells, and error bars represent standard deviation. * *p* < 0.05, ** *p* < 0.01, *** *p* < 0.001; **** *p* < 0.0001.

**Figure 6 cancers-15-00238-f006:**
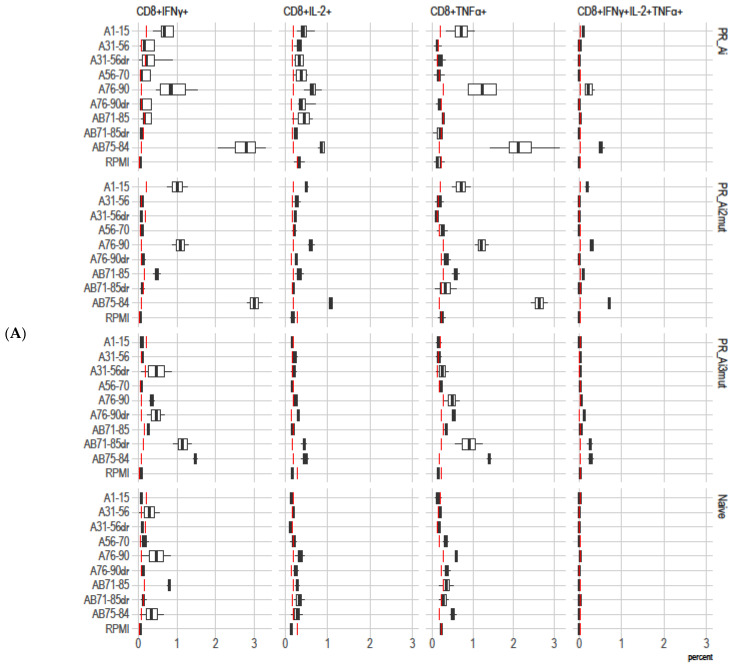
CD8+ T-cell response of mice DNA-immunized with PR_A variants to stimulation with peptides representing immunodominant epitopes of PR assessed by flow cytometry with intracellular cytokine staining (ICCS). BALB/c mice were immunized with plasmids encoding PR_Ai, PR_Ai2mut, PR_Ai3mut, or empty vector pVAX1 (Table 1, series II). Two weeks post-booster-immunization, mice (n = 5 per group) were sacrificed, and their splenocytes were purified and frozen at −80 °C. A week later, cells were thawed and subjected to stimulation with PR-derived peptides (Figure 1C). IFN-γ, IL-2, and TNF-α and triple-cytokine production by stimulated T cells was assessed by flow cytometry with ICCS as described in the Materials and Methods. The frequencies of cytokine-positive CD8+ T-cells secreting IFN-γ, IL-2, or TNF-α, or triple IFN-γ/IL-2/TNF-α cytokine-secreting cells illustrated by box with whiskers; red line illustrates data for pVAX1-immunized mice (**A**); triple-cytokine production upon stimulation with peptides A1–15 (**B**), AB71–85 (**C**), AB71–85dr (**D**); AB75–84 (**E**), A76–90 (**F**), A76–90dr (**G**). Difference between groups was analyzed using a Kruskal–Wallis test, and then pair-wisely by a Mann–Whitney U-test (MW). Groups exhibiting no difference were clustered; resulting clusters were reanalyzed using a Mann–Whitney U-test (Appendix A). KW*, *p* < 0.05; KW (*), *p* < 0.1; MW: * *p* < 0.05, ** *p* < 0.01.

**Figure 7 cancers-15-00238-f007:**
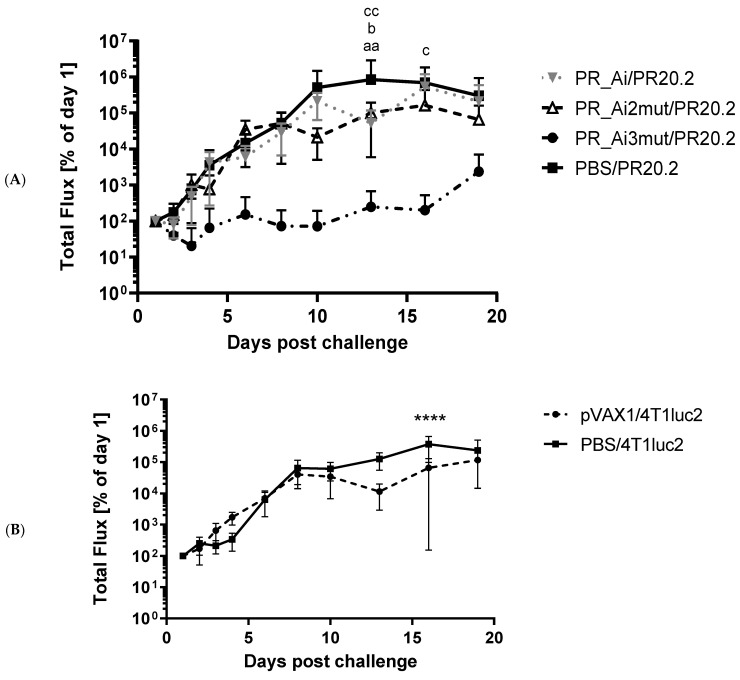
Immunization with PR_Ai3mut identical to PR variant expressed by tumor cells protects mice against tumor challenge, while no protection is rendered by immunization with PR variants different in three (PR_Ai) or one aa residue (PR_Ai2mut). Mice were immunized as depicted in series IV, Table 1, and implanted with 4T1luc2_PR20.2 cells post-boost. Tumor growth was assessed by bioluminescent imaging (BLI) (**A**,**B**), the dynamics of BLI signals in mice immunized with PR_Ai, PR_Ai2mut, and PR_Ai3mut (**A**) and challenged with 4T1luc2_PR20.2 compared to control mice; the dynamics of BLI signals in mice immunized with pVAX1 and challenged with 4T1luc2 compared to control mice in the same cages that received PBS (**B**); Total flux as % from day 1 at 16 (**C**) *p* value < 0.01 was set as significant. Volume of tumors in all groups, each PR-group compared to control mice receiving PBS and challenged with 4T1luc2_PR20.2 (n = 9) and the pVAX1-immunized group compared to control mice receiving PBS and challenged with 4T1luc2 (n = 3) (**D**); Typical images of H&E stainings of high-grade (Grade 3) adenocarcinomas formed by PR20.2 cells after subcutaneous implantations into PBS-immunized mice ((**E**), panels 1–6). Panels depict: (**E1**)—solid spindle cell architecture, (**E2**)—high mitotic activity (areas indicated by red arrows), (**E3**)—weakly preserved glandular structure, (**E4**)—stromal desmoplasia, (**E5**,**E6**)—necrotic areas (red circles). Magnifications ×200 to 400. * *p* < 0.05, **** *p* < 0.0001.

**Figure 8 cancers-15-00238-f008:**
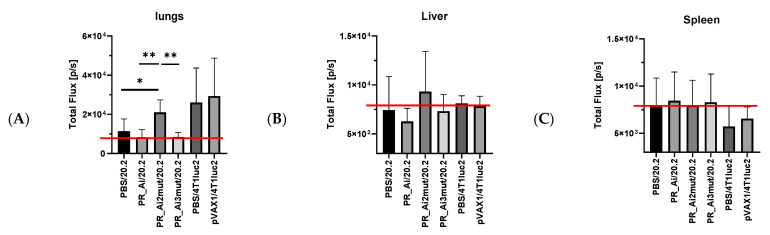
DNA immunization with protease variant PR_Ai3mut identical to PR variant expressed by tumor cells restricted the migration activity of the latter; DNA immunization with the other protease variant had either no (PR_Ai) or a stimulative effect (PR_Ai2mut) on tumor cell migration in vivo. Assessment of cell migration into the lungs (**A**), liver (**B**), and spleen (**C**). Mice were immunized as depicted in Table 1, series IV, and 12 days post-booster-implanted with 4T1luc2_PR20.2 cells. Infiltration of tumor cells into mouse organs was assessed by ex vivo bioluminescent imaging (BLI). Total flux from organs of immunized mice (n = 5 for each immunogen) was compared to that in mice which had received PBS and were challenged with 4T1luc2_PR20.2 cells (n = 9). Control mice were immunized with pVAX1 (n = 5) and challenged with 4T1luc2; control mice are compared to mice from the same cage that received PBS (n = 3). Red line indicates average background level 8000 p/s. Data were analyzed by Kruskal–Wallis followed by pair-wise comparisons by Mann–Whitney U-test. ns- *p* > 0.05, * *p* < 0.05, ** *p* < 0.01.

**Figure 9 cancers-15-00238-f009:**
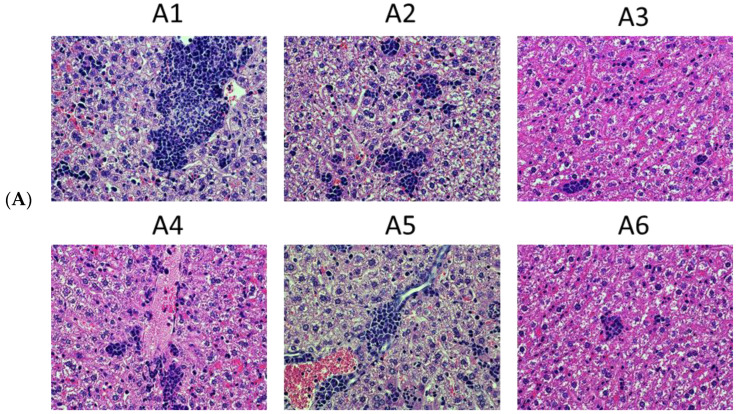
Immunization with PR_Ai3mut identical to the PR variant expressed by tumor cells restricted their metastatic activity of tumor cells/primary tumors in the liver and PR_Ai2mut induced an increase in the number of metastases formed by tumor cells and neutrophils, while no restriction of metastatic activity was observed in mice immunized with the PR_Ai variant. Mice were immunized as depicted in series IV, Table 1, and implanted with 4T1luc2_PR20.2 cells post-boost. Typical images of H&E stainings of liver sections and metastases ((**A**) panels 1–6). Panels depict: (**A1**)—single large metastasis with neutrophils; (**A2**)—metastasis surrounded by multiple micrometastases; (**A3**)—widespread micrometastases; (**A4**)—vessel invasion of metastasis; (**A5**)—infiltration of bile duct; (**A6**)—metastasis consisting of only tumor cells. Magnifications ×200 to 400. Comparisons of number (**B**) and size (**C**) of metastases formed by tumor cells with immune cells between PR_Ai-variants-immunized mice (n = 5 for each immunogen) and PBS-immunized mice (n = 9) challenged with 4T1luc2_PR20.2 cells, and vector-immunized mice (n = 5) and PBS-immunized control (n = 3) challenged with 4T1luc2 cells. Differences were analyzed by ordinary two-way ANOVA with Dunnett’s multiple comparisons correction. ns- *p* > 0.05, * *p* < 0.05, ** *p* < 0.01, *** *p* < 0.001, **** *p* < 0.0001.

**Figure 10 cancers-15-00238-f010:**
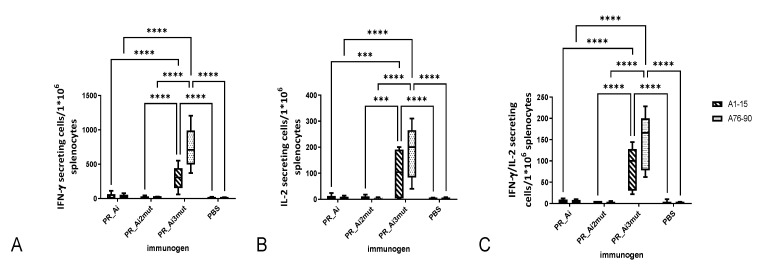
Splenocytes of mice immunized with PR_Ai3mut identical to the PR variant expressed by tumor cells demonstrated a potent IFN-γ/IL-2 immune response to the commonly recognized epitopes A1–15 and A76–90. Mice were immunized as depicted in series IV, Table 1, and implanted with 4T1luc2_PR20.2 cells post-boost. Twenty-one days after challenge, the splenocytes of mice were stimulated in vitro with individual PR-derived peptides (Figure 1C) for 18 h, as described in the Materials and Methods. The in vitro secretion of IFN-γ (**A**) and IL-2 (**B**) and the dual secretion of IFN-γ/IL-2 (**C**) were measured as the number of signal-forming units (sfu) per mln splenocytes; average per group. Differences were analyzed by ordinary two-way ANOVA with Tukey’s multiple comparisons correction. *** *p* < 0.001, **** *p* < 0.0001.

**Table 1 cancers-15-00238-t001:** Scheme of immunization of BALB/c mice to evaluate immunogenicity of PR_A variants.

Series	Group	Nn Mice	DNA Immunogen	Immunization		Tumor Challenge	Immune Response Tests		Immunotoxicity Tests		
				Prime, Dose	Boost, Dose	(Cell Line)	T-Cell Response	Antibody Response	Blood Formula	Blood Biochemistry	Other Tests
I	I-I	5	PR_A	20 ug × 2 sites	none	no	IFN-γ/IL-2 Fluorospot	yes	no	no	no
	I-2	5	PR_Ai								
	I-3	5	PR_B								
	I-4	5	PR_Bi								
	I-5	5	pVAX1								
II	II-1	9	PR_Ai	20 ug × 2 sites	20 ug × 2 sites	no	IFN-γ/IL-2 FluorospotFACS	yes	no	no	no
	II-2	9	PR_Ai2mut								
	II-3	9	PR_Ai3mut								
	II-4	9	pVAX1								
III	III-1	6	PR_Ai	20 ug × 3 sites	20 ug × 3 sites	no	no	no	Appendix A	Appendix A	Appendix A
	III-2	6	PR_Ai2mut + PR_Ai3mut (1:1)								
	III-3	6	pVAX1								
	III-4	6	PBS								
IV	IV-1	5	PR_Ai	20 ug × 2 sites	20 ug × 2 sites	4T1luc2_PRAi3mut (PR20.2)	IFN-γ/IL-2 Fluorospot	no	no	no	no
	IV-2	5	PR_Ai2mut								
	IV-3	5	PR_Ai3mut								
	IV-4	9	PBS								
	IV-5	5	pVAX1			4T1luc2					
	IV-6	3	PBS								

**Table 2 cancers-15-00238-t002:** Characteristics of the immune status of mice DNA receiving the plasmid encoding the inactivated consensus protease of HIV-1 FSU_A strain (PR_Ai) or an equimolar mix of plasmids encoding PR_Ai variants with the DR mutations PR_Ai2mut and PR_Ai3mut (DR PR mix) compared to control mice receiving pVAX1 or PBS, assessed 1 and 12 days post-plasmid-boost (Table 1, series III).

Immunogen	PR_Ai *		PR DR mix *		PR_Ai **		PR DR Mix **		pVAX1 **		PBS	
Day of Assessment Post-Boost	Day 1	Day 12	Day 1	Day 12	Day 1	Day 12	Day 1	Day 12	Day 1	Day 12	Day 1	Day 12
Thymus % (AVE ± stdv)	0.13 ± 0.03	0.17 ± 0.03	0.14 ± 0.04	0.12 ± 0.05	0.13 ± 0.03	0.17 ± 0.03	0.14 ± 0.04	0.12 ± 0.05	0.11 ± 0.03	0.12 ± 0.02	0.16 ± 0.03	0.10 ± 0.04
Stats, Thymus % ***	ns *	* p * = 0.017	ns	ns	ns	* p * = 0.004	ns	ns	* p * = 0.03	ns	N/A	N/A
Spleen % (AVE ± stdv)	0.49 ± 0.09	0.44 ± 0.03	0.39 ± 0.06	0.38 ± 0.04	0.49 ± 0.09	0.44 ± 0.03	0.39 ± 0.06	0.38 ± 0.04	0.39 ± 0.03	0.48 ± 0.13	0.43 ± 0.03	0.44 ± 0.02
Stats, Spleen % ***	ns	ns	ns	ns	ns	ns	ns	* p * = 0.008	ns	ns	N/A	N/A
Axil LN (AVE ± stdv)	0.02 ± 0.01	0.02 ± 0.01	0.01 ± 0.01	0.02 ± 0.01	0.02 ± 0.01	0.02 ± 0.01	0.01 ± 0.01	0.02 ± 0.01	0.04 ± 0.02	0.02 ± 0.01	0.04 ± 0.02	0.02 ± 0.01
Stats, Axil LN ***	ns	ns	* p * = 0.009	ns	ns	ns	ns	ns	ns	ns	N/A	N/A
WBC (AVE ± stdv)	16.22 ± 3.41	12.70 ± 2.44	9.42 ± 3.95	7.38 ± 2.56	16.22 ± 3.41	12.70 ± 2.44	9.42 ± 3.95	7.38 ± 2.56	10.80 ± 3.54	11.90 ± 1.62	8.97 ± 1.09	9.68 ± 3.13
Stats, WBC ***	* p * = 0.026	ns	ns	* p * = 0.009	* p * = 0.002	ns	ns	ns	ns	ns	N/A	N/A
LYM (AVE ± stdv)	63.97 ± 15.96	70.60 ± 2.33	70.92 ± 7.75	74.50 ± 1.33	63.97 ± 15.96	70.60 ± 2.33	70.92 ± 7.75	74.50 ± 1.33	71.53 ± 2.91	71.50 ± 4.24	63.27 ± 5.02	66.82 ± 7.02
Stats, LYM ***	ns	ns	ns	ns	ns	ns	* p * = 0.04	* p * = 0.008	* p * = 0.01	ns	N/A	N/A
GRAN (AVE ± stdv)	33.57 ± 15.90	25.52 ± 3.00	26.72 ± 15.89	22.80 ± 1.13	33.57 ± 15.90	25.52 ± 3.00	26.72 ± 15.89	22.80 ± 1.13	25.75 ± 2.68	26.04 ± 4.09	43.02 ± 4.92	30.45 ± 6.39
Stats, GRAN ***	ns	ns	ns	ns	ns	ns	* p * = 0.04	* p * = 0.004	* p * = 0.01	ns	N/A	N/A
RBC (AVE ± stdv)	8.90 ± 0.23	10.00 ± 0.20	9.20 ± 1.11	9.74 ± 0.53	8.90 ± 0.23	10.00 ± 0.20	9.20 ± 1.11	9.74 ± 0.53	9.66 ± 0.20	10.10 ± 0.32	8.90 ± 0.09	9.86 ± 0.30
Stats, RBC ***	* p * = 0.002	ns	ns	ns	ns	ns	ns	ns	* p * = 0.004	ns	N/A	N/A
PLT (AVE ± stdv)	918.17 ± 47.72	807.50 ± 180.74	950.17 ± 63.10	806.17 ± 239.41	918.17 ± 47.72	807.50 ± 180.74	950.17 ± 63.10	806.17 ± 239.41	1066.83 ± 63.55	712.20 ± 177.18	899.17 ± 128.71	874.83 ± 158.26
Stats, PLT ***	* p * = 0.004	ns	* p * = 0.016	ns	ns	ns	ns	ns	* p * = 0.016	ns	N/A	N/A
Metamyelocytes (AVE ± stdv)	5.00 ± 1.41	4.00 ± 0.63	5.17 ± 0.41	4.00 ± 0.63	5.00 ± 1.41	4.00 ± 0.63	5.17 ± 0.41	4.00 ± 0.63	5.50 ± 0.55	3.40 ± 0.55	5.67 ± 0.52	3.83 ± 1.17
Stats, Metamyelocytes ***	* p * = 0.015	ns	ns	ns	ns	ns	ns	ns	ns	ns	N/A	N/A

* Compared to pVAX1; ** compared to PBS; *** Statistic was carried out with a Kruskal–Wallis test with Dunn’s multiple comparisons correction. red letter indicate significant changes.

**Table 3 cancers-15-00238-t003:** Parameters of the biochemical blood analysis in mice receiving the plasmid encoding the inactivated consensus protease of HIV-1 FSU_A strain (PR_Ai) or an equimolar mix of plasmids encoding PR_Ai variants with the DR mutations PR_Ai2mut and PR_Ai3mut (DR PR mix) compared to control mice receiving pVAX1 or PBS, assessed 1 and 12 days post-plasmid-boost.

Immunogen	PR_Ai *		PR DR mix *		PR_Ai **		PR DR Mix **		pVAX1 **		PBS	
Assessment Day	Day 1	Day 12	Day 1	Day 12	Day 1	Day 12	Day 1	Day 12	Day 1	Day 12	Day 1	Day 12
HCT (AVE ± stdv)	40.50 ± 0.55	46.17 ± 0.94	42.33 ± 5.61	45.23 ± 1.77	40.50 ± 0.55	46.17 ± 0.94	42.33 ± 5.61	45.23 ± 1.77	44.50 ± 1.05	44.98 ± 2.57	40.83 ± 0.75	45.77 ± 1.47
HCT	* p * = 0.002	ns	ns	ns	ns	ns	ns	ns	* p * = 0.04	ns	N/A	N/A
HGB (AVE ± stdv)	128.00 ± 2.19	145.17 ± 2.64	133.17 ± 17.14	142.00 ± 7.54	128.00 ± 2.19	145.17 ± 2.64	133.17 ± 17.14	142.00 ± 7.54	144.20 ± 8.04	139.83 ± 2.99	129.67 ± 2.25	147.67 ± 5.50
HGB	* p * = 0.002	ns	ns	ns	ns	ns	ns	ns	* p * = 0.004	ns	N/A	N/A
Total protein (AVE ± stdv)	55.76 ± 17.62	67.72 ± 10.10	67.88 ± 3.06	65.44 ± 16.41	55.76 ± 17.62	67.72 ± 10.10	67.88 ± 3.06	65.44 ± 16.41	81.00 ± 12.72	56.54 ± 7.76	64.28 ± 12.75	63.00 ± 12.81
Total protein	ns	ns	* p * = 0.016	ns	ns	ns	ns	ns	ns	ns	N/A	N/A
ALT (AVE ± stdv)	51.66 ± 22.64	35.16 ± 4.27	35.22 ± 16.53	29.04 ± 5.30	51.66 ± 22.64	35.16 ± 4.27	35.22 ± 16.53	29.04 ± 5.30	31.98 ± 7.51	46.20 ± 13.06	40.38 ± 8.78	41.70 ± 14.70
ALT	ns	ns	ns	* p * = 0.03	ns	ns	ns	ns	ns	ns	N/A	N/A
Urea (AVE ± stdv)	4.51 ± 0.44	7.71 ± 1.57	4.76 ± 1.83	5.32 ± 0.96	4.51 ± 0.44	7.71 ± 1.57	4.76 ± 1.83	5.32 ± 0.96	4.56 ± 0.87	8.93 ± 2.57	5.65 ± 1.20	8.93 ± 2.57
Urea	ns	ns	ns	* p * = 0.03	ns	ns	ns	* p * = 0.03	* p * = 0.009	ns	N/A	N/A
TGC (AVE ± stdv)	0.98 ± 0.23	1.39 ± 0.23	0.94 ± 0.11	1.73 ± 0.24	0.98 ± 0.23	1.39 ± 0.23	0.94 ± 0.11	1.73 ± 0.24	0.74 ± 0.05	1.45 ± 0.28	1.26 ± 0.21	1.26 ± 0.24
TGC	ns	ns	* p * = 0.008	ns	ns	ns	* p * = 0.028	* p * = 0.03	* p * = 0.046	ns	N/A	N/A
Cholesterol (AVE ± stdv)	1.82 ± 0.61	2.39 ± 0.21	2.36 ± 0.29	2.20 ± 0.43	1.82 ± 0.61	2.39 ± 0.21	2.36 ± 0.29	2.20 ± 0.43	2.58 ± 0.27	2.16 ± 0.56	2.09 ± 0.35	2.13 ± 0.31
Cholesterol	* p * = 0.03	ns	ns	ns	ns	ns	ns	ns	* p * = 0.016	ns	N/A	N/A

* Compared to pVAX1; ** compared to PBS.

**Table 4 cancers-15-00238-t004:** Inverse correlation of tumor size with cellular immune response to immunodominant epitopes of consensus HIV-1 FSU_A protease (PR_A) represented by synthetic peptides encompassing PR aa 1–15 (A1–15) and aa 76–90 (A76–90) assessed by a dual IFN-γ/IL-2 Fluorospot of splenocytes of DNA-immunized and PR20.2-challenged mice (Table 1, series IV) stimulated with respective peptides in vitro at the experimental endpoint.

	Tumor Volume Correlated with					
	A1–15			A76–90		
	IFN-γ	IL-2	IFN-γ/IL-2	IFN-γ	IL-2	IFN-γ/IL-2
Spearman r	−0.4202	−0.3659	−0.6293	−0.4147	−0.4229	−0.6121
95% confidence interval	−0.7106 to −0.007482	−0.6773 to 0.05673	−0.8277 to −0.2911	−0.7073 to −0.0008411	−0.7122 to −0.01072	−0.8187 to −0.2654
*p* (two-tailed)	0.0409	0.0787	0.0010	0.0439	0.0395	0.0015

## Data Availability

Not applicable.

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
