# Peer review of "HIV-1 Protease as DNA Immunogen against Drug Resistance in HIV-1 Infection: DNA Immunization with Drug Resistant HIV-1 Protease Protects Mice from Challenge with Protease-Expressing Cells"

_cancers, 2022, doi:10.3390/cancers15010238_

Round 1
Reviewer 1 Report
The manuscript by Stefan et al describes HIV-1 protease as DNA immunogen against drug resistance in HIV-1 infection and immunization with drug resistant HIV-1 protease protects mice from tumor challenge. The manuscript is well written containing detailed experiments and results with appropriate controls supporting the conclusion. Furthermore, it has implication in developing immunotherapies against drug resistant HIV-1 infection. However, the following important points must be addressed before consideration.
1. The authors need to explain convincingly the significance and the new findings of their studies compared to previous reports that DNA immunization with HIV-1 protease (PR) is advanced for immunotherapy of HIV-1 infection to reduce number of infected cells producing drug resistant virus.
2. The Title needs a modification. At a face value DNA immunization with drug resistant HIV-1 protease protects mice from tumor challenge implies it can be applied for cancer treatment.
3. The mice tumor challenge study is just an assay to demonstrate cellular immune responses to a tumor cell line expressing the HIV antigen. Therefore, it is puzzling that this manuscript is submitted to a cancer journal. An appropriate submission would be to HIV or other Viruses related journals.
4. The challenge study is performed using a mouse model with no human cells present. HIV only infects human cells and the immune system as well as antigen recognition mechanisms would be significantly different between human and mouse models where there are no human cells present. Therefore, the relevance of the study and conclusion could be misleading.
Author Response
Reviewer 1: The manuscript by Stefan et al describes HIV-1 protease as DNA immunogen against drug resistance in HIV-1 infection and immunization with drug resistant HIV-1 protease protects mice from tumor challenge. The manuscript is well written containing detailed experiments and results with appropriate controls supporting the conclusion. Furthermore, it has implication in developing immunotherapies against drug resistant HIV-1 infection. However, the following important points must be addressed before consideration.
Response: Dear Reviewer, thank you very much for the thorough review of our manuscript, positive evaluation and critical comments, which we have addressed below.
Comment 1: The authors need to explain convincingly the significance and the new findings of their studies compared to previous reports that DNA immunization with HIV-1 protease (PR) is advanced for immunotherapy of HIV-1 infection to reduce number of infected cells producing drug resistant virus.
Response: Our group had been developing approaches to vaccinate against drug resistance on HIV infection for over twenty years (DOI: 10.1016/j.vaccine.2003.10.052; 10.1016/j.vaccine.2005.08.020; 10.1159/000053996; 10.1038/s41598-018-26281-z). Protease is one of the logical targets of such vaccines. Andreas Bråve, David Hällengard developed a candidate PR based vaccine based on PR gene of HIV-1 HXB2 strain into which they have introduced the most common DR mutations V82F, I84V (https://doi.org/10.1016/j.vaccine.2010.10.083). We have cited their studies. We have used PR inactivation they applied to achieve high level of PR expression in eukaryotic cells, but developed this approach further. Firstly, we designed consensus gene of PR of HIV-1 clade A FSU strain that infects millions of people on the territory of former Soviet Union and which has so far remained highly conserved, at least in the genes encoding HIV enzymes, due to rapid transfer within IDU population. This makes our vaccine candidate applicable for prevention of DR development in a large group of population. Secondly, we introduced mutations characteristic and most common to this strain, not a set of commonly seen mutations in PR. This makes our vaccine candidate relevant to epidemiological situation in the region of HIV-1 FSU_A circulation. Thirdly, and most importantly, we have developed a functional model to test the efficacy of PR based candidate vaccine in mice by creating murine tumor cells expressing HIV-1 PR and challenging vaccinated mice with these cells to mimic HIV-1 infection (infection with cells, not free virus transfer). This model allowed to demonstrate the protective capacity of PR DNA vaccination, which was not shown earlier.
Comment 2: The Title needs a modification. At a face value DNA immunization with drug resistant HIV-1 protease protects mice from tumor challenge implies it can be applied for cancer treatment.
Response: The title has been changed to «HIV-1 protease as DNA immunogen against drug resistance in HIV-1 infection: DNA immunization with drug resistant HIV-1 protease protects mice from challenge with protease expressing tumor cells»
Comment 3: The mice tumor challenge study is just an assay to demonstrate cellular immune responses to a tumor cell line expressing the HIV antigen. Therefore, it is puzzling that this manuscript is submitted to a cancer journal. An appropriate submission would be to HIV or other Viruses related journals.
Response: The topic of this manuscript is not directly related to cancer. Manuscript was submitted to the Special issue covering the international symposium “Chronic Viral Infections and Cancer, Openings for Vaccines” held online on December 16-17, 2021 and sponsored by CANCERS. For reference, see Isaguliants MG, Trotsenko I, Buonaguro FM. An overview of "Chronic viral infection and cancer, openings for vaccines" virtual symposium of the TechVac Network - December 16-17, 2021. Infect Agent Cancer. 2022 Jul 8;17(Suppl 2):28. doi: 10.1186/s13027-022-00436-0. We would also like to pinpoint that the study demonstrates capacity of a DNA vaccine to prevent growth and metastatic activity of highly aggressive tumor cells, and also reveals that the protective potential of such vaccines may depend on cellular immune response against single epitopes within tumor antigen. This issue is important for cancer vaccine development, as has been indicated in the Conclusions.
Comment 4: The challenge study is performed using a mouse model with no human cells present. HIV only infects human cells and the immune system as well as antigen recognition mechanisms would be significantly different between human and mouse models where there are no human cells present. Therefore, the relevance of the study and conclusion could be misleading.
Response: Authors thank the Reviewer for highlighting this aspect. Current challenge model does not contain any human cells, it is based on immunocompetent animals and syngenic cell lines expressing viral antigens. Models analogues to this are widely used to assess efficacy of newly generated vaccines, one of the best examples being therapeutic vaccines against HPV (https://doi.org/10.1186/2045-3701-4-11; 10.1134/S0026893322050028). We have followed a standard procedure for vaccine design and immunogenicity testing, plus developed a model for vaccine efficacy testing to evaluate its protectivity in a mouse model. Further experiments in humanized mice, or in animals which can be infected with HIV-1, or SIV, or SHIV, can only be performed after obtaining the proof of efficacy in the basic models, such as mice.
Importantly, even the very first next step after the mouse model, inclusion of human cell of any origin would require the use of immune compromised animals. One can use hu-PBL-SCID immunocompromised mice with engrafted human peripheral blood lymphocytes for challenge. We have reviewed such models and found that the quality of immune response to immunization and tumor challenge and predicative value of this response to further human studies are not very high (doi: 10.1134/S0026893322050028). Besides, the use of such models would require special conditions for animal care, and lead to an increased cost of experiments. At the same time, we fully agree with the Reviewer that further tests of PR based DNA immunogens against DR in HIV infection are needed, and that the clinical outcome of vaccination with drug resistant HIV-1 protease can only be fully evaluated in the clinical trials.
Reviewer 2 Report
DNA/RNA-based vaccines represent a breakthrough in the generation of effective vaccines. Petkov et al. focus on the development of a therapeutic vaccine against HIV-1 which would induce a lasting control of HIV-1 infection. They developed a candidate vaccine against drug resistance HIV-1 based on the consensus sequence of PR of HIV-1, and evaluated its immunogenicity, immunotoxicity, and protective potential in mice by challenging immunized animals with syngenic PR-expressing tumor cells mimicking viral infection. I would like to congratulate the researchers for their very good work, the wide variety of methods they use to prove their research, and their precise description.
I don't see any line markings in the manuscript, which makes it a little difficult to find my comments.
The introduction is well written and provides enough information to acquaint readers with the problem and current developments on the subject. This also applies to all other chapters of the manuscript.
I have a few notes in the Materials and Methods chapter:
Page 3: I expected to see the transformation of E. coli described already in Design and generation of consensus FSU-A protease genes. I see that you described it under expression of the respective plasmids.
Alternatively, staining was done with monoclonal antibody 1696 (Exbio; 1:2000) _Please, specified the antibody.
PR – please give information about the expected molecular weight of recombinant PR proteins.
Nowhere in the WB do I see the use of a molecular marker (or more precisely its graphical transfer from SDS PAGE). Please add molecular markers of WB and indicate the molecular weight of PR.
Page 7 -Humoral immune responses assessed by ELISA – probably need to be italic
Primary antibody binding was detected using secondary polyclonal goat anti-mouse IgG HRP conjugate (Dako, Denmark) diluted 1:2000 in Scan buffer. –It is not clear here what it is primary Ab (mice serum)?
Assessment of T cell immune response by Fluorospot and flow cytometry – Italic
Regarding the methodology used in this scientific development, I can only congratulate you.
Page 10. Please insert Figure 1 as close as possible after its first citation.
Page 14 Earlier studies demonstrated that single V82A mutation causes up to two-fold decrease in the enzyme activity ([57] and references therein), however, mutations at aa positions 46 and 54 compensate for the loss [58], which could explain similar activity of all three PR_A variants. To conclude, synthetic PR_A genes encoded proteins of expected molecular mass, had enzymatic activity characteristic to HIV-1 protease, and were stained with PRspecific antibodies.
My point is that this paragraph belongs to the discussion, not the results.
3.4.1. PR_A and PR_Ai variants are strongly immunogenic after single DNA immunization - Italic. I see the rest of your bullet points are not in Italics either, is that a Cancer requirement?
Figure 4 is multi-component, wouldn't it be better to split it into 2 figures to make it easier for readers to follow? Please label the individual panels (A, B, C, E) in the text so that it is easy for the reader to find their way around. This also applies to your other multi-component figures, please consider how to present them so that they are easier for readers to understand. Some of the information may be transferred to the Supplementary Materials.
From these data, we concluded that (i) introduction of inactivation mutation D25N increased the immunogenicity of PR_A as was shown before for PR_B [41]; (ii) all inactivated PR_A variants, both with and without DR mutations, were immunogenic on the cellular level, mostly for CD8+ T cells; (iii) DNA immunization with DR PR_Ai variants induced cellular immune response recognizing peptides bearing V82A mutation, not recognized by mice DNA immunized with the parental PR_A; (iv) PR encoding plasmids induced no antibody response. Is this for the result or for discussion?
In general, you have so many results that you can easily present them in two manuscripts.
The discussion is well-written and provides an opportunity to assess the contributions of this study.
Author Response
Reviewer 2: DNA/RNA-based vaccines represent a breakthrough in the generation of effective vaccines. Petkov et al. focus on the development of a therapeutic vaccine against HIV-1 which would induce a lasting control of HIV-1 infection. They developed a candidate vaccine against drug resistance HIV-1 based on the consensus sequence of PR of HIV-1, and evaluated its immunogenicity, immunotoxicity, and protective potential in mice by challenging immunized animals with syngenic PR-expressing tumor cells mimicking viral infection. I would like to congratulate the researchers for their very good work, the wide variety of methods they use to prove their research, and their precise description.
Response: Dear Reviewer, thank you very much for the thorough review and for high appreciation of our manuscript, and all critical comments. Revision of the manuscript following these points helped us to considerably improve it.
Comment 1: I don't see any line markings in the manuscript, which makes it a little difficult to find my comments.
Response: Unfortunately, MDPI template does not allow us to ad line markings in the manuscript, however we tracked all the corrections so one can easily track all the changes done during the revision.
Comment 2: Page 3: I expected to see the transformation of E. coli described already in Design and generation of consensus FSU-A protease genes. I see that you described it under expression of the respective plasmids.
Response: In the section Design and generation of consensus FSU-A protease genes we described only the cloning of designed genes into the vectors for pro- and eukaryotic expression. Purification of plasmids were performed from E.coli with plasmid purification kits as recommended by the manufacturer. Transformation of E.coli, mentioned briefly in the section “Recombinant HIV-1 proteases and protease-specific antibodies”, was also mentioned as the first step of detailed protocol for protein production and purification. Detailed description of the standard well known protocol of bacterial transformation was omitted in an attempt to limit the length of the already long manuscript.
Comment 3: Alternatively, staining was done with monoclonal antibody 1696 (Exbio; 1:2000) _Please, specified the antibody.
Response: Description of the antibody used was modified as follows: staining was done with mouse monoclonal antibody clone 1696 (Exbio; cat 11-302, 1:2000)
Comment 4: PR – please give information about the expected molecular weight of recombinant PR proteins.
Response: Results, page 14. Information was added as follows: “The recombinant protein with expected molecular mass of 11 kDa was recognized by the polyclonal antibodies directed...”
Comment 5: Nowhere in the WB do I see the use of a molecular marker (or more precisely its graphical transfer from SDS PAGE). Please add molecular markers of WB and indicate the molecular weight of PR.
Response: During all the experiments molecular marker “Page ruler prestained” (Thermofisher) was used. This ladder has line corresponding to 10 and 15kDa. BAnd, corresponding to 10 kDa marker was added to all WB figures. Incorportating a band corresponding to 15kDa lane would lead to an empty space on top of the image and grossly enlarge the figure, therefore, we have chosen not to add it.
Comment 6: Page 7 -Humoral immune responses assessed by ELISA – probably need to be italic
Response: Section title was made italic, and text font was changed similar to the main text font.
Comment 7: Primary antibody binding was detected using secondary polyclonal goat anti-mouse IgG HRP conjugate (Dako, Denmark) diluted 1:2000 in Scan buffer. –It is not clear here what it is primary Ab (mice serum)?
Response: Text was modified as follows: “mouse serum reactivity with PR was detected using secondary polyclonal goat anti-mouse IgG HRP conjugate (Dako, Denmark) diluted 1:2000 in Scan buffer.”
Comment 8: Assessment of T cell immune response by Fluorospot and flow cytometry – Italic
Response: Section title was made italic, and text font was changed similar to the main text font.
Comment 9: Page 10. Please insert Figure 1 as close as possible after its first citation.
Response: Figure 1 was transferred below the first paragraph containing it’s citation.
Comment 10: Page 14 Earlier studies demonstrated that single V82A mutation causes up to two-fold decrease in the enzyme activity ([57] and references therein), however, mutations at aa positions 46 and 54 compensate for the loss [58], which could explain similar activity of all three PR_A variants. To conclude, synthetic PR_A genes encoded proteins of expected molecular mass, had enzymatic activity characteristic to HIV-1 protease, and were stained with PRspecific antibodies.
My point is that this paragraph belongs to the discussion, not the results.
Response: We agree that this sentence contains elements of the discussion of our results. However, to make discussion as clear as possible, we focused it on PR expression in eukaryotic cells as it could and does affect the immunologenicity of respective DNA immunogen. The sentence mention above described that plasmids we designed encoded proteins with predictive enzymatic activity and can be adequately used as DNA immunogens. Text referred to by the Reviewer carries this message. We feel that transferring it to discussion will make discussion too heterogeneous and complicated, and therefore have chosen to leave this text in the Results section.
Comment 11: 3.4.1. PR_A and PR_Ai variants are strongly immunogenic after single DNA immunization - Italic. I see the rest of your bullet points are not in Italics either, is that a Cancer requirement?
Response: Format of the section title was made by CANCERS Editorial office with section of the first order (i.e. Results) marked in bold, second order (i.e. section 3.4. Immunogenic performance of plasmids encoding PR_A variants in mice) marked in italic and third order (i.e. 3.4.1. PR_A and PR_Ai variants are strongly immunogenic after single DNA immunization) in format of the main text.
Comment 12: Figure 4 is multi-component, wouldn't it be better to split it into 2 figures to make it easier for readers to follow? Please label the individual panels (A, B, C, E) in the text so that it is easy for the reader to find their way around. This also applies to your other multi-component figures, please consider how to present them so that they are easier for readers to understand. Some of the information may be transferred to the Supplementary Materials.
Response: Indeed, figures representing immune response data are especially multi-component, however they could not be split as panel A in Figure 4 and panels A, B in Figure 5 represent pile-ups of total immune response and following panels illustrate different aspects/parameters of specific immune response constituting the pile-up. Each observation refer to a specific panel of the multi-component figure. In our view, they are better presented together in one figure.
Comment 13: From these data, we concluded that (i) introduction of inactivation mutation D25N increased the immunogenicity of PR_A as was shown before for PR_B [41]; (ii) all inactivated PR_A variants, both with and without DR mutations, were immunogenic on the cellular level, mostly for CD8+ T cells; (iii) DNA immunization with DR PR_Ai variants induced cellular immune response recognizing peptides bearing V82A mutation, not recognized by mice DNA immunized with the parental PR_A; (iv) PR encoding plasmids induced no antibody response. Is this for the result or for discussion?
Response: After a discussion, the co-authors came to conclusion that this text contains a summary, or rather, a list of results, without elements of discussion, and agreed to retain it in the Results section.